# Layer specific and general requirements for ERK/MAPK signaling in the developing neocortex

Lei Xing[1], Rylan S Larsen[2†], George Reed Bjorklund[3], Xiaoyan Li[1], Yaohong Wu[1], Benjamin D Philpot[1,4,5], William D Snider[1,4,5], Jason M Newbern[3*]

[1]University of North Carolina Neuroscience Center, The University of North Carolina School of Medicine, Chapel Hill, United States; [2]Allen Institute for Brain Science, Seattle, United States; [3]School of Life Sciences, Arizona State University, Tempe, United States; [4]Department of Cell Biology and Physiology, University of North Carolina at Chapel Hill, Chapel Hill, United States; [5]Carolina Institute for Developmental Disabilities, The University of North Carolina School of Medicine, Chapel Hill, United States

**Abstract** Aberrant signaling through the Raf/MEK/ERK (ERK/MAPK) pathway causes pathology in a family of neurodevelopmental disorders known as 'RASopathies' and is implicated in autism pathogenesis. Here, we have determined the functions of ERK/MAPK signaling in developing neocortical excitatory neurons. Our data reveal a critical requirement for ERK/MAPK signaling in the morphological development and survival of large Ctip2[+] neurons in layer 5. Loss of *Map2k1/2 (Mek1/2)* led to deficits in corticospinal tract formation and subsequent corticospinal neuron apoptosis. ERK/MAPK hyperactivation also led to reduced corticospinal axon elongation, but was associated with enhanced arborization. ERK/MAPK signaling was dispensable for axonal outgrowth of layer 2/3 callosal neurons. However, *Map2k1/2* deletion led to reduced expression of Arc and enhanced intrinsic excitability in both layers 2/3 and 5, in addition to imbalanced synaptic excitation and inhibition. These data demonstrate selective requirements for ERK/MAPK signaling in layer 5 circuit development and general effects on cortical pyramidal neuron excitability.

*For correspondence: jason.
newbern@asu.edu

Present address: [†]Allen Institute for Brain Science, Seattle, United States

Competing interests: The authors declare that no competing interests exist.

## Introduction

The canonical Ras/Raf/MEK/ERK (ERK/MAPK) signaling pathway is a key intracellular signaling cascade downstream of cell surface receptors critical for brain development (*Samuels et al., 2009*). In the developing cortex, the ERK/MAPK pathway is thought to be particularly important for neuronal responses to neurotransmitters and receptor tyrosine kinase (RTK) ligands, such as FGFs and neurotrophins. In the mature brain, it is well established that ERK/MAPK plays a central role in the activity-dependent plasticity of neural circuits (*Shilyansky et al., 2010*; *Thomas and Huganir, 2004*).

Importantly, a number of human neurodevelopmental syndromes have been linked to aberrant ERK/MAPK activity. This related group of human syndromes, increasingly referred to as RASopathies, are caused by genetic mutations in core components or regulators of the ERK/MAPK signaling cascade (*Rauen, 2013*). Macrocephaly, neurodevelopmental delay, cognitive impairment, and epilepsy are frequently observed in RASopathy patients with clinical manifestations being dependent on the precise causative mutation (*Rauen, 2013*). RASopathies are most often associated with hyperactive ERK/MAPK signaling (eg. Neurofibromatosis type 1 (NF1), Noonan, Costello, and Cardiofaciocutaneous (CFC) syndromes) (*Rauen, 2013*). However, mutations that lead to diminished ERK/MAPK activation have been identified in a subset of LEOPARD and CFC syndrome patients

**eLife digest** In the nervous system, cells called neurons form networks that relay information in the form of electrical signals around the brain and the rest of the body. Typically, an electrical signal travels from branch-like structures at one end of the cell, through the cell body and then along a long fiber called an axon to reach junctions with another neurons.

The connections between neurons start to form as the nervous system develops in the embryo, and any errors or delays in this process can cause severe neurological disorders and intellectual disabilities. For example, genetic mutations affecting a communication system within cells known as the ERK/MAPK pathway can lead to a family of syndromes called the "RASopathies". Abnormalities in this pathway may also contribute to certain types of autism. However, it is not clear how alterations to the ERK/MAPK pathway cause these conditions.

Xing et al. investigated whether ERK/MAPK signaling regulates the formation of connections between neurons and the activity of neurons in mouse brains. The experiments showed that the growth of axons that extend from an area of the brain called the cerebral cortex towards the spinal cord are particularly sensitive to changes in the level of signaling through the ERK/MAPK pathway. On the other hand, inhibiting the pathway has relatively little effect on the growth of axons within the cerebral cortex. Further experiments showed that many neurons in the cerebral cortex require the ERK/MAPK pathway to activate genes that alter neuronal activity and the strength of the connections between neurons.

Xing et al.'s findings suggest that defects in the connections between the cerebral cortex and different regions of the nervous system may contribute to the symptoms observed in patients with conditions linked to alterations in ERK/MAPK activity. Future studies will focus on understanding the molecular mechanisms by which ERK/MAPK pathway influences the organization and activity of neuron circuits during the development of the nervous system.

(*Kontaridis et al., 2006*; *Nowaczyk et al., 2014*). Abnormal Ras/MAPK signaling has also been observed in models of other monogenic neurodevelopmental disorders including Fragile X syndrome and Tuberous Sclerosis (*Chévere-Torres et al., 2012*; *Faridar et al., 2014*; *Osterweil et al., 2013*; *Zhang et al., 2014*). Recent exciting work has shown that pharmacological normalization of pathological Ras/MAPK activity is sufficient for correcting select cellular and behavioral abnormalities in Fragile X, Tuberous Sclerosis, NF1, Noonan, and Costello syndrome mutant mice (*Cui et al., 2008*; *Lee et al., 2014*; *Li et al., 2005*; *Osterweil et al., 2013*; *Wang et al., 2012*; *Zhang et al., 2014*). However, our understanding of these disorders remains rudimentary as there is limited information on brain cell type-specific consequences of either loss- or gain-of-function through this pathway.

Accumulating evidence also suggests that pathological ERK/MAPK signaling contributes to certain forms of autism. Altered Ras/MAPK signaling has been identified as a common downstream mediator of divergent genetic mutations linked to autism, and *MAPK3/ERK1* is present in a region of 16p11.2 mutated in ~1% of cases of autism (*Eichler and Zimmerman, 2008*; *Gilman et al., 2012*; *Gilman et al., 2011*; *Kumar et al., 2008*; *Pinto et al., 2010*; *Pucilowska et al., 2015*; *Weiss et al., 2008*). Little is known about how ERK/MAPK signaling might relate to the pathogenesis of autism. An important current research theme is that the behavioral manifestations of autism spectrum disorders (ASDs) may be linked to both functional hypo- and hyper-connectivity between distinct brain regions (*Geschwind and Levitt, 2007*; *Just et al., 2007*; *Keown et al., 2013*; *Supekar et al., 2013*). Furthermore, recent work in postmortem brains of autistic patients suggests that local patches of disorganization, in which cortical layers 4–5 are particularly affected, play an important role in disease pathogenesis (*Stoner et al., 2014*). In one study, co-expression network analyses of autism-linked genetic mutations suggested that layer 5 in prefrontal and sensorimotor cortex is a key site of convergence for pathogenesis (*Willsey et al., 2013*). Whether aberrant ERK/MAPK signaling might result in cortical layer disorganization and defective long-range connectivity is unknown.

To address questions of cell type specificity and consequences for circuit formation, we have defined the effects of ERK/MAPK loss- and gain-of-function on the development of cortical

pyramidal neurons. Pyramidal neuron-specific functions of ERK/MAPK signaling were assessed by deleting the upstream kinases *Map2k1/Mek1* and *Map2k2/Mek2* (hereafter referred to as *Map2k1/2*) or overexpressing hyperactive *Map2k1*. Conditional deletion of *Map2k1/2* led to major disruption of layer 5 with noticeably fewer CTIP2-expressing large neurons compared to controls. Further, long range axon extension of layer 5 corticospinal projection neurons during early development was markedly impaired. Subsequent to delayed entry of axons into the cervical spinal cord, many layer 5 projection neurons in sensorimotor cortices underwent apoptosis. Gain-of-function ERK/MAPK signaling also affected layer 5 CST neurons with a resultant decrease in axon elongation and associated increase in axon branching. The morphological requirement for ERK/MAPK signaling was specific for layer 5, as layer 2/3 was not disrupted and callosal projection neurons in upper cortical layers do not exhibit overt changes in axon extension or targeting following *Map2k1/2* deletion. In contrast to the layer-specific functions of ERK/MAPK on axonal development, we found that ERK/MAPK was required for the expression of ARC and other plasticity-associated genes across all cortical lamina. Further, loss of ERK/MAPK signaling in pyramidal neurons disrupted excitatory and inhibitory neurotransmission and altered intrinsic excitability in both layers 2/3 and 5. Our data reveal unexpectedly specific requirements for ERK/MAPK signaling in layer 5 circuit development and general effects on the excitability of cortical pyramidal neurons in multiple layers.

## Results

### Excitatory neuron-specific modification of ERK/MAPK activity

Previous work has shown that MAPK1/3 (aka ERK1/2) is activated in embryonic cortical neurons, albeit at much lower levels than in the ventricular zone (*Faedo et al., 2010*; *Li et al., 2014*; *Pucilowska et al., 2012*; *Toyoda et al., 2010*). In western blots of sensorimotor cortical lysates from P1, 2, 7, 14, and 21 day old mice, we found that the levels of pan-MAPK1/3(ERK1/2) and pan-MAP2K1/2 show a steady but evident increase from a relatively lower level at birth (*Figure 1—figure supplement 1A*). Phosphorylated-MAPK1/3(ERK1/2) and phosphorylated-MAP2K1/2 levels were also relatively low at birth but increased noticeably by P7 and peaked at P14 (*Figure 1—figure supplement 1A*) (*Oliveira et al., 2008*). The pattern of phosphorylated-MAPK1/3(ERK1/2) expression in P3 histological sections of cortex did not exhibit any clear-cut laminar specificity (*Figure 1—figure supplement 1B*). These findings indicate that ERK/MAPK signaling is activated in the developing cortex, peaking during the second postnatal week.

We generated a mouse model to test the direct, neuron-autonomous role of ERK/MAPK signaling by inactivating *Map2k1/2* specifically in immature mouse cortical excitatory neurons. We conditionally deleted *Map2k1/2* with a Cre-dependent *Map2k1* allele, a germ-line *Map2k2* deletion allele, and Cre-recombinase under the control of the *NeuroD6/Nex* promoter (*Map2k1^{loxp/loxp}*;*Map2k2^{-/-}*; *Neurod6-Cre*,referred to hereafter as *Map2k1/2;Neurod6-Cre*) (*Goebbels et al., 2006*). As expected, *Neurod6-Cre* activated reporter-gene expression in the Cre-dependent EYFP mouse line, *Ai3*, in excitatory, but not inhibitory, neurons in the neocortex by mid-embryogenesis (*Figure 1—figure supplement 1C–E*) (*Madisen et al., 2010*). Western blotting of neocortical lysates and immunolabeling show that *Map2k1/2;Neurod6-Cre* mice exhibit significantly reduced MAP2K1 levels by birth and reduced phosphorylation of ERK/MAPK substrates, RSK, and MSK (*Figure 1—figure supplement 1F–H*). Complete loss of MAP2K1 protein would not be expected in whole cortical lysates due to MAP2K1 expression in inhibitory interneurons and non-neuronal cell types (*Figure 1—figure supplement 1F*). These data show that the *Neurod6-Cre*-mediated genetic targeting strategy is effective at inducing loss of ERK/MAPK signaling in developing cortical excitatory neurons.

*Map2k1/2;Neurod6-Cre* pups were born at normal Mendelian ratios without overt differences from littermate controls. However, a delay in overall growth could be detected by the end of the first postnatal week and lethality was invariably observed in the third to fourth postnatal week. Behaviorally, P14 *Map2k1/2;Neurod6-Cre* mice exhibited spontaneous and persistent hindlimb clasping when lifted by the tail, an indicator of neurological impairment. A qualitatively similar effect on growth, neurological function, and viability was also observed in *Mapk1^{loxp/loxp}*;*Mapk3^{-/-}*;*Neurod6-Cre* mice (data not shown), demonstrating that phenotypes are conserved following deletion of different core components of the ERK/MAPK cascade.

*Map2k1/2;Neurod6-Cre* mice exhibited an average reduction in body weight of 37.1 ± 11.3% at P14 and neocortical volume was reduced by 23.2 ± 6.4% (mean ± SEM, n=7, <0.0001). A significant increase in neuron (NEUN/RBFOX3$^+$ cell) density was observed in *Map2k1/2;Neurod6-Cre* sensory cortices from 1636 ± 185 neurons/mm$^2$ to 2168 ± 231 neurons/mm$^2$ (mean ± SEM, n=5, p=0.002) (*Figure 1A–B*). Given the significant difference in cortical size, we extrapolated a relative estimate of neuronal number by multiplying the density measurement by the relative change in cortical volume between individual littermate control and mutant pairs. This estimate showed no significant difference in relative global neuron number between mutant and control cortices (98.0 ± 3.5%, mean ± SEM, n=5, p=0.61). We did, however, detect a significant reduction in the size of NEUN$^+$ soma in P14 *Map2k1/2;Neurod6-Cre* sensory cortices in all cortical layers with the most pronounced effect on layer 5 neurons (*Figure 1C*). In contrast, no decrease in the size of parvalbumin-expressing, GABAergic neurons in the cortex was observed (data not shown). These data show that ERK/MAPK signaling is necessary for neurological function and the somal growth of excitatory neurons, but do not provide strong support for a widespread role in regulating global excitatory neuron number.

Recombination in *Neurod6-Cre* mice is particularly strong in the cerebral cortex, however, sparse recombination has been detected outside of the cortex (*Figure 1—figure supplement 1C*) (*Goebbels et al., 2006*). To confirm the deficits in *Map2k1/2;Neurod6-Cre* mice are specifically due to cortical malfunction, we deleted *Map2k1/2* using an independent Cre line, *Emx1-Cre. Emx1-Cre* induces recombination in dorsal telencephalic progenitors by E9.5 that generate excitatory neurons specifically within the cortex and hippocampus (*Gorski et al., 2002*). *Map2k1$^{loxp/loxp}$;Map2k2$^{-/-}$; Emx1-Cre (Map2k1/2;Emx1-Cre)* exhibit cortex specific MAP2K1 deletion and a pattern of growth delay, clasping behavior, and lethality similar to that observed in *Map2k1/2;Neurod6-Cre* mice (*Figure 1—figure supplement 1I–J*). We conclude that ERK/MAPK signaling in cortical excitatory neurons is necessary for appropriate nervous system function and mouse viability.

## Disruption of layer 5 and reduced number of CTIP2$^+$ neurons after ERK/MAPK inactivation

The profound decrease in layer 5 neuron soma size caused by *Map2k1/2* deletion suggests that ERK/MAPK signaling is particularly important for the development of this layer. Comprehensive analyses of layer 5 neurons have identified specific gene expression patterns, morphologies, and electrophysiological characteristics for this neuronal subtype (*Greig et al., 2013*; *Kwan et al., 2012*; *Leone et al., 2008*). We analyzed the expression pattern of a critical transcription factor for layer 5 development, CTIP2/BCL11B, in control and mutant cortices (*Figure 1A–B*) (*Arlotta et al., 2005*). The appearance of co-labeled CTIP2$^+$/NEUN$^+$ neurons in layer 5 appeared disrupted in rostral P14 *Map2k1/2;Neurod6-Cre* cortices compared to controls (*Figure 1B*, yellow arrows).

CTIP2$^+$ layer 5 neurons are a small percentage of the total cortical neuron population. In the sensory cortex of control mice, we found that CTIP2$^+$ layer 5 neurons represent only 9.61 ± 0.84% (mean ± SEM, n=5) of all NEUN$^+$ neurons within a radial column. Thus, our previous global NEUN estimates across an entire cortical column were not sensitive enough for detecting changes confined to sparse neuronal subtypes. To quantify the number of CTIP2$^+$ neurons, we determined the relative proportion of cells in layer 5 that express CTIP2$^+$ as a percentage of NEUN$^+$ cells across all lamina in a radial cortical column. This measurement was performed in motor, sensory, and visual cortices. Indeed, we found that P14 *Map2k1/2;Neurod6-Cre* primary motor and sensory cortices exhibit a clear decrease in the relative proportion of CTIP2$^+$ neurons in layer 5 (*Figure 1D–G, J*). Qualitatively similar results were observed in P14 *Map2k1/2;Emx1-Cre* cortices (*Figure 2—figure supplement 1A–B*). The relative proportion of CTIP2$^+$ neurons in visual cortices (*Figure 1H–I, J*) was not significantly diminished in *Map2k1/2;Neurod6-Cre* mutants, suggesting the effect in sensory and motor cortex was due to loss of the cell type and not a result of ERK/MAPK regulation of global CTIP2 expression levels.

## Failure of CST development and absence of large tufted neurons in layer 5

Layer 5 neurons are morphologically heterogeneous with distinct subpopulations that can be differentiated by cortico-cortical or subcortical axonal projections. We examined layer 5 projections in the hindbrain corticobulbar tract (CBT) and spinal cord corticospinal tract (CST) of P14 *Map2k1/2* mutant

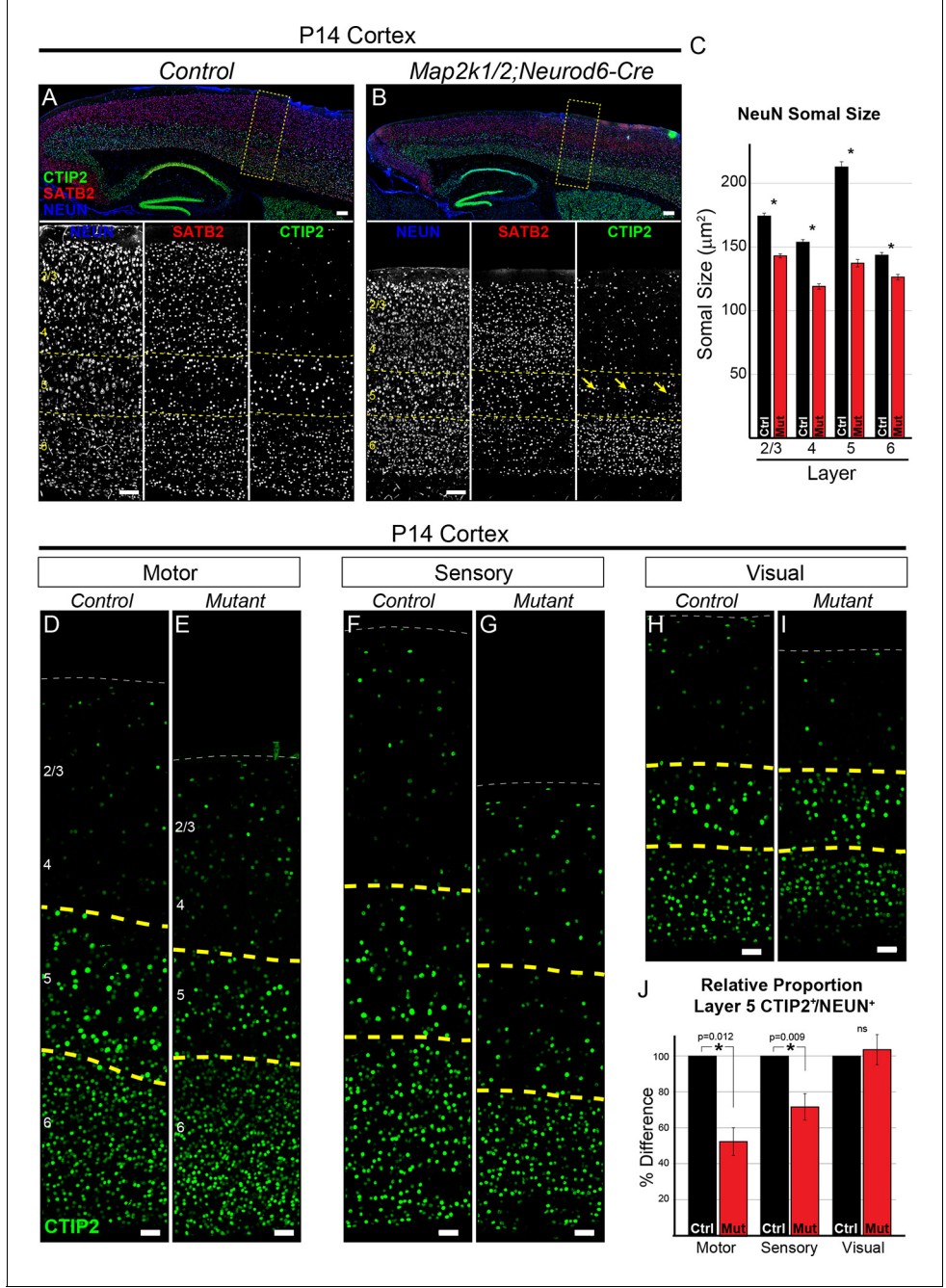

**Figure 1.** Loss of ERK/MAPK signaling leads to a reduction in the number of CTIP[+] layer 5 neurons and reduced neuronal somal size. (A–B). Immunostaining of P14 control (A) and *Map2k1/2;Neurod6-Cre* (B) sagittal forebrain sections for all neurons, callosal projection neurons, and subcortical projection neurons with NEUN, SATB2, and CTIP2, respectively, revealed an aberrant pattern of CTIP2 expression in layer 5 of mutant sensorimotor cortices (yellow arrows) (n=6, scale bar=100 µm). (C) Quantification of somal size was performed by measuring the cross-sectional area of randomly selected, NEUN labeled soma in distinct cortical layers. A reduction in the size of NEUN labeled soma in P14 mutant sensory cortices was detected in all cortical layers, but was particularly pronounced in layer 5 (n= >300 total neurons per layer derived from five independent mice per condition, mean ± SEM, *p<0.001). (D–I) Representative confocal images of CTIP2 immunolabeling in radial columns of primary motor (D–E), sensory (F–G), and visual (H–I) cortex from P14 control (D, F, H) and *Map2k1/2;Neurod6-Cre* (E, G, I) brains (scale bar=30 µm). (J) Quantification of the relative number of layer 5 CTIP2[+] neurons as a proportion of the total number of NEUN[+] neurons in a cortical column revealed a substantial decrease in motor and sensory, but not visual, cortices in P14 mutant mice (n=4, mean ± SEM, *p<0.05).

The following figure supplement is available for figure 1:

**Figure supplement 1.** Developmental changes in ERK/MAPK activity and mouse models for loss of ERK/MAPK signaling in cortical excitatory neurons.

mice. Immunostaining for a well-established marker of corticospinal projections, PKCγ, and genetic labeling with *Ai3* revealed a profound decrease in the size of the CBT in P14 *Map2k1/2;Neurod6-Cre* hindbrains (*Figure 2A–B*, *Figure 2—figure supplement 1E–F*). CST labeling was also strikingly reduced in the cervical spinal cord and essentially absent in the lumbar spinal cord in both *Map2k1/2;Neurod6-Cre* (*Figure 2C–D*) and *Map2k1/2;Emx1-Cre* mutants (*Figure 2—figure supplement 1C–D*). Analysis of rare *Map2k1/2* mutants that survived as late as P24 revealed that no CST axons were present in lumbar spinal cords (data not shown). These data provide further evidence of an overt loss of corticospinal neurons by the second postnatal week.

Corticospinal neurons represent a subset of the entire CTIP2$^+$ population in sensorimotor layer 5 (*Arlotta et al., 2005*). The lack of layer 5 neuron loss in the *Map2k1/2;Neurod6-Cre* visual cortex suggests that ERK/MAPK signaling is dispensable for the development of projection neurons targeting structures other than the spinal cord. Past work has shown that callosally projecting layer 5 neurons have significantly shorter and less complex apical dendritic arbors than subcortical projection neurons (*Larsen et al., 2007*; *Molnár and Cheung, 2006*). To determine which of these classes was affected by *Map2k1/2* deletion, a Thy1-based reporter, *YFP-16*, that fluorescently labels a small proportion of layer 5 neurons in sensorimotor cortex, was bred with *Map2k1/2;Neurod6-Cre* mice (*Feng et al., 2000*). These mutants clearly show that the fluorescently labeled layer 5 neurons in P14-P21 *Map2k1/2;Neurod6-Cre;YFP-16* sensorimotor cortices have substantially shorter apical dendrites than in controls (*Figure 2E–F*). The reduction in large, tufted neurons in layer 5 of mutant mice provide further evidence for a deficit in the development of corticospinal projection neurons.

## Gene expression profiling of ERK/MAPK inactivated cortices

To gain further insight into the cellular and molecular mechanisms utilized by ERK/MAPK signaling to regulate cortical development, we performed microarray profiling of P14 whole cortical lysates from control and *Map2k1/2;Neurod6-Cre* mutants. The validity of the screen was verified by a highly significant -2.76 ± 0.13 (p-val<0.001)-fold reduction in the expression of *Map2k1*-exon 3 at the probe-level in P14 mutant samples. We observed a pronounced decrease in a large number of genes that are known to be highly expressed in layer 5 projection neurons at P9 or P14, including *Etv1/Er81*, *Adcyap1*, *Ldb2*, *Dkk3*, *Lmo4*, and *Opn3* (fold-change>1.5 p<0.05) (*Arlotta et al., 2005*; *Chen et al., 2005*; *Chen et al., 2005*; *Molyneaux et al., 2007*) (*Figure 2—source data 1*). Moreover, *Fezf2*, a well-known master transcription factor important for corticospinal neuron development, showed a strong trend toward diminished expression in *Map2k1/2;Neurod6-Cre* mutants (*Chen et al., 2005*; *Chen et al., 2005*; *Molyneaux et al., 2005*). The gene profiling results provide further support for a specific loss of layer 5 neurons possibly due to cell death or altered fate specification in *Map2k1/2* mutant mice.

## ERK/MAPK signaling is necessary for corticospinal axon extension

The loss of layer 5 neurons following deletion of *Map2k1/2* could be due to an early disruption in the initial specification of this neuronal subtype. During layer 5 neuron development, *Neurod6-Cre* is not expressed until neurons are post-mitotic (*Goebbels et al., 2006*; *Wu et al., 2005*). However, it remained possible that post-mitotic stages of embryonic layer 5 neuron specification were altered during embryogenesis. In newborn mutant pups, we found that the expression and number of CTIP2$^+$ neurons in presumptive layer 5 was not diminished following *Map2k1/2* deletion (*Figure 3A–C*). Thus, ERK/MAPK signaling is not required for the initial establishment of the correct numbers of the CTIP2$^+$ deep-layer neuron population.

By P3 in wild-type neonates, corticospinal axons have projected through the ventral hindbrain, crossed the midline at the medullary/spinal cord boundary, and are extending into cervical spinal cord through the dorsal funiculus (*Schreyer and Jones, 1982*). We asked whether ERK/MAPK signaling was required for the initial outgrowth of corticospinal projections in vivo. In vivo DiI injections into the motor cortex of P0.5 neonates were performed to assess subcortical axon growth, especially into the spinal cord (*Figure 3—figure supplement 1A*). Strikingly, analysis of anterogradely labeled axonal projections at P3-4 revealed a highly significant decrease in the extension of corticospinal axons into the lower cervical/upper thoracic segments of spinal cord (*Figure 3D–E*). Moreover, a profound decrease in the caudal extension of descending corticospinal axons into the spinal cord of P2 *Map2k1/2;Emx1-Cre;Ai3* mice was also observed (*Figure 3F,H*). DiI, PKCγ, and Ai3 labeling of

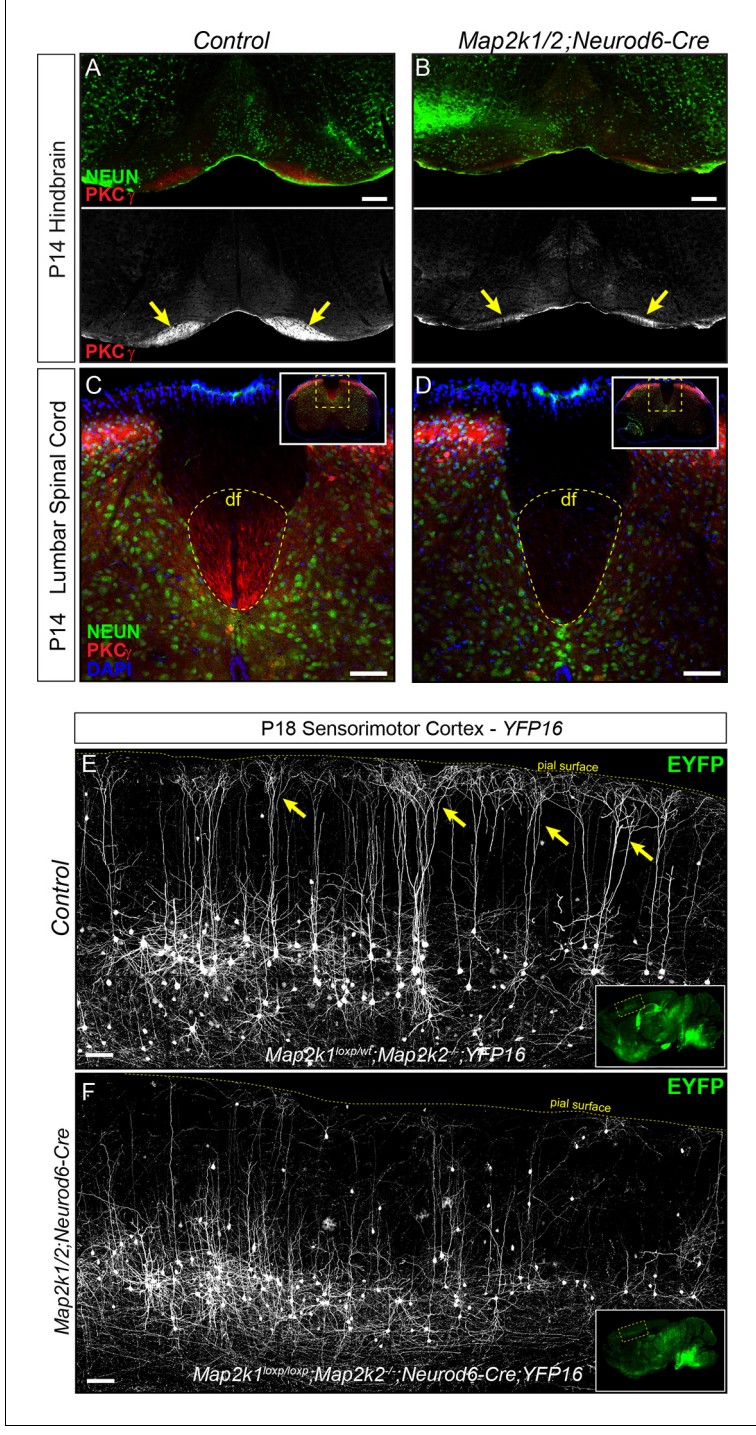

**Figure 2.** Corticospinal tract defects in *Map2k1/2;Neurod6-Cre* mice. (**A–D**) To further evaluate the loss of layer 5 projection neurons, we examined the expression of a well-established corticospinal tract marker, PKCγ. Compared to control hindbrains (**A**) and spinal cords (**C**), a profound decrease in corticospinal tract labeling was observed in the *Map2k1/2;Neurod6-Cre* hindbrain (B-yellow arrows, scale bar=200 μm) and spinal cord (**D**), consistent with the reduced number of CTIP2[+] layer 5 neurons (df=dorsal funiculus, n=3, scale bar=50 μm). (**E–F**) The Thy1-YFP reporter line, *YFP16*, labels a small fraction of layer 5 neurons in sensorimotor cortex. Many layer 5 neurons labeled in this line have large complex apical dendritic arbors (yellow arrows) that are consistent with the known morphology of subcerebral projection neurons, including corticospinal neurons. EYFP expressing neurons heavily branch in layer 1 as shown in representative confocal images of sagittal sections from P18 control mice (**E**, yellow

*Figure 2 continued on next page*

*Figure 2 continued*
arrows). In *Map2k1/2;Neurod6-Cre* cortices, we noted a significant reduction in the number of EYFP labeled neurons in layer 5 with complex apical dendritic arbors (**F**) (n=3, scale bar=100 μm).
The following source data and figure supplement are available for figure 2:
**Source data 1.** Reduced expression of layer 5 neuron markers following loss of *Map2k1/2.*
**Figure supplement 1.** Reduced number of corticospinal neurons following loss of *Map2k1/2.*

subcortical projections in *Map2k1/2;Neurod6-Cre* and *Map2k1/2;Emx1-Cre* mice revealed that layer 5 neuron growth into the tectum and pons/medulla was not significantly decreased (*Figure 3G,I*; *Figure 3—figure supplement 1A–D*). These data demonstrate that the initial growth of corticospinal axons to the level of the hindbrain is not significantly disrupted in *Map2k1/2* mutants, however, ERK/MAPK signaling is clearly necessary for corticospinal axon elongation into the spinal cord.

In vitro studies have demonstrated that IGF1 signaling acting via PI3K and ERK/MAPK promotes corticospinal axon outgrowth in vitro (*Özdinler and Macklis, 2006*). IGF1 is also focally and intensely expressed in the medulla in the presumptive inferior olivary nucleus during the precise developmental time frame (P1 – P2) that CST axons normally enter the spinal cord (*Figure 3—figure supplement 1E*). IGF1 expression is temporally regulated, being present by E18 and much diminished by P14 (data not shown). *Map2k1/2* deletion in CST axons does not affect IGF1 expression by these cells (*Figure 3—figure supplement 1E–F*). Thus, IGF1 may act via ERK/MAPK within corticospinal neurons to regulate corticospinal outgrowth.

We functionally tested whether cortical neuron specific loss of IGF1R would disrupt the CST development by generating *Igf1r<sup>loxp/loxp</sup>;Neurod6-Cre* mutants (*Liu et al., 2009*). Total body weight and brain size were significantly reduced in *Igf1r<sup>loxp/loxp</sup>;Neurod6-Cre* mutants (*Figure 3—figure supplement 1I*) and a clear loss of IGF1R immunoreactivity was observed in P7 *Igf1r<sup>loxp/loxp</sup>;Neurod6-Cre* cortices (*Figure 3—figure supplement 1G–H*). However, the expression of CTIP2 in layer 5 was not significantly altered (*Figure 3—figure supplement 1J–K*). Further, PKCγ labeling of the CST in the cervical and lumbar spinal cord did not reveal a significant difference in rostrocaudal extension in *Igf1r<sup>loxp/loxp</sup>;Neurod6-Cre* mutants (*Figure 3J–P*). These data suggest that direct regulation of ERK/MAPK in layer 5 neurons by IGF1R signaling is not a significant regulator of neuronal survival and CST development in vivo.

## Caspase-3 activation in Layer 5 neurons after ERK/MAPK deletion

By P3 we noted a dramatic increase in the number of activated caspase-3 labeled neurons in layer 5 in *Map2k1/2;Neurod6-Cre* mutants when compared to controls (*Figure 4A–C*). Modest caspase-3 activation was also observed in layer 6, but not in upper layers 1–4 (*Figure 4C*). In line with the elevation of caspase-3 activity, we observed IBA1<sup>+</sup> microglia in P3 *Map2k1/2;Neurod6-Cre* cortices with a ramified morphology that appeared to be engulfing CTIP2<sup>+</sup> neurons (*Figure 4D–E*). To further confirm the absence of corticospinal neurons, we retrogradely labeled corticospinal projection neurons by injecting DiI into the cervical spinal cord in *Map2k1/2;Neurod6-Cre* neonates at P3, the time point when dying cells could initially be detected in the cortex. Analysis of retrogradely labeled corticospinal neurons in the primary motor cortex at P7 revealed a significantly reduced number of DiI labeled cells consistent with the death of these neurons during this time period (*Figure 4F–G*). Interestingly, retrograde DiI labeling of P5 layer 5 neurons that project into the contralateral hemisphere did not reveal a substantial decrease in P9 mutants relative to controls (*Figure 4H–I*). These data indicate that loss of *Map2k1/2* results in overt caspase-3 activation and corticospinal neuron death that can be first detected at P3-4.

We examined the possibility that transformation of CTIP2<sup>+</sup> corticospinal neurons into an alternative phenotype during the neonatal period contributes to the reduction in CTIP2<sup>+</sup> layer 5 neuron number observed at P14. We tested whether CTIP2<sup>+</sup> layer 5 neurons might be transforming into a callosal phenotype by quantifying the proportion of cortical neurons that express SATB2, an important transcription factor for the differentiation of callosal neurons (*Alcamo et al., 2008*; *Britanova et al., 2008*). Since SATB2 is coexpressed by a minority of CTIP2<sup>+</sup> neurons and SATB2

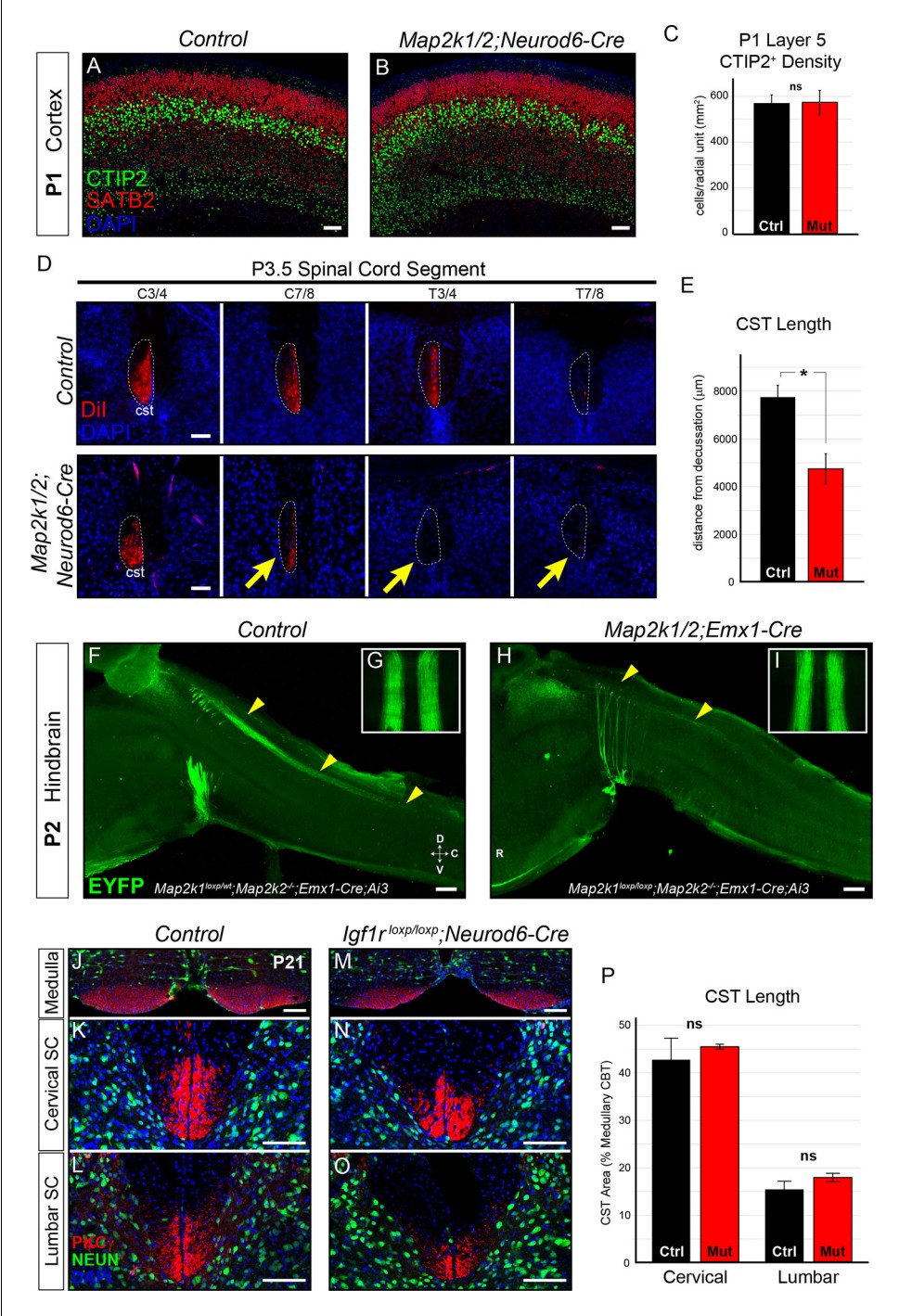

**Figure 3.** Layer 5 neuron corticospinal axon outgrowth requires ERK/MAPK signaling in vivo. (A–C) Expression of a well-known master transcription factor for layer 5 neurons, CTIP2, was intact in cortical layer 5 as shown in representative confocal images of newborn control (A) and *Map2k1/2; Neurod6-Cre* (B) sensorimotor cortices (scale bar=50 µm). Quantitation of the number of CTIP2-expressing nuclei in cortical layer 5 (C) did not reveal a significant difference in CTIP2+ neuron density between control and mutant neonates (n=3, mean ± SEM, p=0.89). (D–E) In vivo DiI injections into the sensorimotor cortex of P0.5 control and *Map2k1/2;Neurod6-Cre* neonates were performed and mice were collected three days after injection. The extent of anterograde DiI labeling was analyzed in coronal sections through the spinal cord (D). We observed a significant decrease in the extent of corticospinal (cst) elongation in mutant mice, especially in the lower cervical/thoracic spinal cord segments (yellow arrows in D) (n=3, scale bar=100 µm). Quantitation of CST length relative to the medullary decussation revealed a significant decrease in corticospinal axon growth in mutant spinal cords (E) (n=3, mean ± SEM, *p=0.02). (F–I) The initial stages of corticospinal elongation in the dorsal spinal cord can be visualized in sagittal sections of the caudal medulla and rostral spinal cord from P2 *Emx1-Cre;Ai3* mice. Immunoenhancement of the *Ai3* reporter with an EGFP antibody clearly shows that

*Figure 3 continued on next page*

*Figure 3 continued*

the caudal extension of corticospinal axons (yellow arrowheads) is profoundly reduced in P2 *Map2k1^loxp/loxp;Map2k2^-/-;Emx1-Cre;Ai3* mutant mice (H) when compared to *Map2k1^loxp/wt;Map2k2^-/-;Emx1-Cre;Ai3* controls (F) (n=3, scale bar=200 μm). Whole mount visualization of the CST coursing through the ventral medulla in control (G) and mutant (I) hindbrains did not reveal an overt difference in corticospinal growth (n=3). (J–P) Compared to control mice (J–L), *Neurod6-Cre* mediated deletion of *Igf1r* did not alter the relative area of PKCγ labeling in the cervical (K, N, P) or lumbar (L, O, P) CST compared to the medullary CBT (J, M) in mutant mice (M–O) at P21 (n=3, scale bar=50 μm).

The following figure supplement is available for figure 3:

**Figure supplement 1.** The rostrocaudal elongation of layer 5 axons in the spinal cord requires ERK/MAPK, but not IGF1R, signaling.

plays a transient role in corticospinal differentiation, we only assayed SATB2$^+$ layer 5 neurons that were CTIP2$^-$ (*Leone et al., 2015*; *McKenna et al., 2015*). A significant decrease in the proportion of SATB2$^+$/CTIP2$^-$ neurons in layer 5 could be detected in motor cortex, but not sensory cortex, in P14 *Map2k1/2;Neurod6-Cre* mice when compared to controls (*Figure 4J*). These findings demonstrate that the reduced number of CTIP2$^+$ neurons in layer 5 does not coincide with a compensatory increase in the proportion of SATB2$^+$/CTIP2$^-$, presumably callosal, layer 5 neurons.

## Gain-of-function ERK/MAPK signaling reduces CST elongation, but enhances branching

Many neurodevelopmental syndromes that involve mutations in canonical members of the ERK/MAPK cascade exhibit enhanced ERK/MAPK activity (*Rauen, 2013*). Thus, we assessed effects of hyper-activation of the ERK/MAPK pathway on developing cortical excitatory circuits. A Cre-dependent constitutively-active *Map2k1^S217/222Q* (ca-Map2k1) overexpressing line was crossed with *Neurod6-Cre* and *Emx1-Cre* mice to induce gain-of-function ERK/MAPK signaling (*Krenz et al., 2008*). Immunolabeling for MAP2K1 confirmed overexpression of the *ca-Map2k1* allele in the *ca-Map2k1;Neurod6-Cre* cortex (*Figure 5—figure supplement 1*). In contrast to the loss-of-function mutants, *ca-Map2k1;Neurod6-Cre* mice are viable and able to breed, but are reduced in weight. Adult male control mice weighed 41.01 ± 1.17g while mutants were 27.68 ± 0.67g (mean ± SEM, n=14 controls, 11 mutants). *Ca-Map2k1;Emx1-Cre* are viable and grossly normal, but exhibit lethality between 6 and 10 weeks of age.

We first asked whether ERK/MAPK hyperactivation led to defects in lamination, particularly in layer 5. CTIP2 labeling of mature sensorimotor cortices showed no overt defects in the specification or number of layer 5 neurons in *ca-Map2k1;Neurod6-Cre* mice (*Figure 5A–C*). Based on our previous results, we hypothesized that hyperactivation of ERK/MAPK would lead to enhanced corticospinal axon growth into the spinal cord. In vivo DiI injections of P0.5 motor cortices were performed to assess the extent and pattern of layer 5 subcortical axon outgrowth. Innervation of the thalamus and medulla appeared normal in P3.5 gain of function mutants (*Figure 5D*). Surprisingly, we detected a marked decrease in the initial extension of DiI labeled corticospinal afferents into the spinal cord in P3-4 *ca-Map2k1;Neurod6-Cre* mutants (*Figure 5D–E*). Further, genetic labeling of corticospinal projections revealed a significantly diminished CST size in the spinal cord dorsal funiculus of both *ca-Map2k1;Neurod6-Cre* and *ca-Map2k1;Emx1-Cre* mutants (*Figure 6—figure supplement 1A–C*). These data show that enhanced ERK/MAPK signaling results in a decrease in the elongation of corticospinal axons into the spinal cord that persists into adulthood.

We tested whether the final pattern of axonal elongation and arborization was altered in *ca-Map2k1;Neurod6-Cre* mutants using a Cre-dependent, *tdTomato*-expressing viral vector (AAV5-CAG-FLEX-tdTomato). Following unilateral injection of AAV into primary motor cortex at P1, the extent of axonal labeling was assessed in the white matter tract and grey matter in hindbrain (*Figure 6A–F*) and spinal cord (*Figure 6G–J*) sections. To determine the amount of axonal elongation that occurred between the hindbrain and the spinal cord, high resolution confocal imaging and measurement of axonal RFP labeling in the white matter of the hindbrain corticobulbar tract (CBT, *Figure 6C–D*) and cervical spinal cord CST (*Figure 6I–J*) was performed. We then calculated the amount of RFP labeling in the spinal cord white matter tract relative to the amount of labeling in the hindbrain white matter tract within individual mice to provide a measure of axonal elongation. While control mice showed little reduction in the extent of labeling in the spinal cord white matter tract

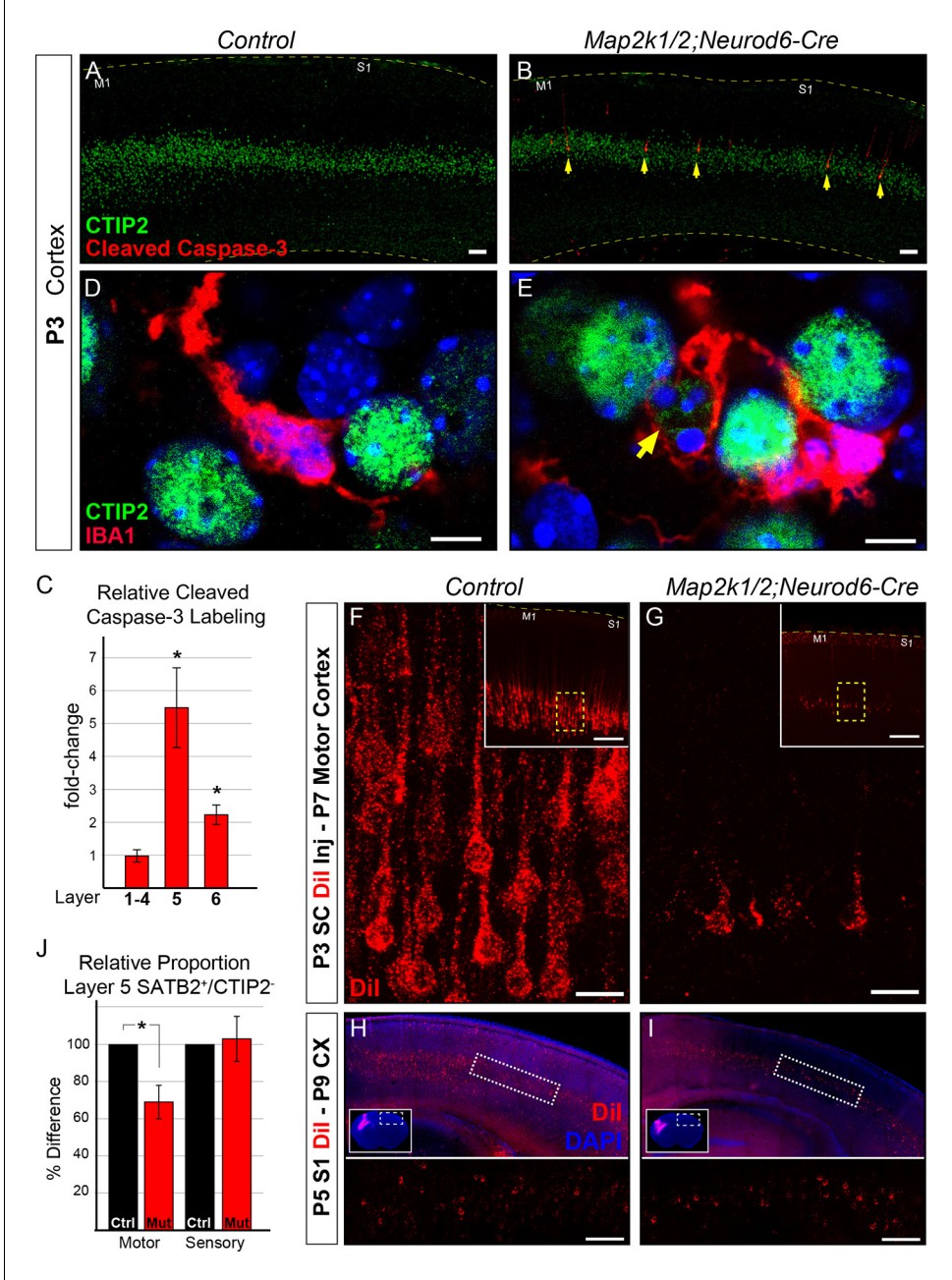

**Figure 4.** Initiation of layer 5 neuron death by P3 in *Map2k1/2;Neurod6-Cre* mutants. (**A–B**) Representative confocal images of immunolabeling for cleaved activated caspase-3, a well-known marker of neuronal apoptosis in P3 control (**A**) and *Map2k1/2;Neurod6-Cre* (**B**) sensorimotor cortices. Note the extensive increase in the number of activated caspase-3+ cells co-labeled with CTIP2 in layer 5 of mutant cortices (B, yellow arrows) (n=4, scale bar=50 µm). (**C**) Quantification of activated caspase-3+ cells in upper layers (layer 1–4), layer 5 (CTIP2+) and layer 6 revealed a pronounced elevation in the number of apoptotic cells in layer 5 in P3 *Map2k1/2;Neurod6-Cre* mice relative to controls. The numbers of activated caspase-3+ cells are comparable in upper layers and doubled in layer 6 relative to controls. (n=4, mean ± SEM, * p<0.05) D-E. Relative to controls (**D**), many microglia (IBA1+) were observed with processes surrounding CTIP2 labeled neurons in layer 5 of mutant cortices (**E**) (n=3, scale bar=5 µm). F-G. Retrograde labeling of corticospinal neurons at P3 via DiI injection into the cervical spinal cord revealed a substantial reduction in the number of DiI labeled neuronal bodies in mutant (**G**) sensorimotor cortices at P7 when compared to controls (**F**) (n=3, scale bar=20 µm). (**H–I**) Retrograde labeling of layer 5 neurons that project into the contralateral hemisphere by injection of DiI in contralateral cortices, did not reveal an obvious loss in mutant (**I**) mice when compared to controls (**H**) (n=3, scale bar=100 um). (**J**) Quantification of the relative number of layer 5 SATB2+/CTIP2- neurons as a proportion of the total number of NEUN+ neurons in a cortical column revealed a substantial decrease in motor, but not sensory, cortices in mutant mice (n=3, mean ± SEM, *p=0.003).

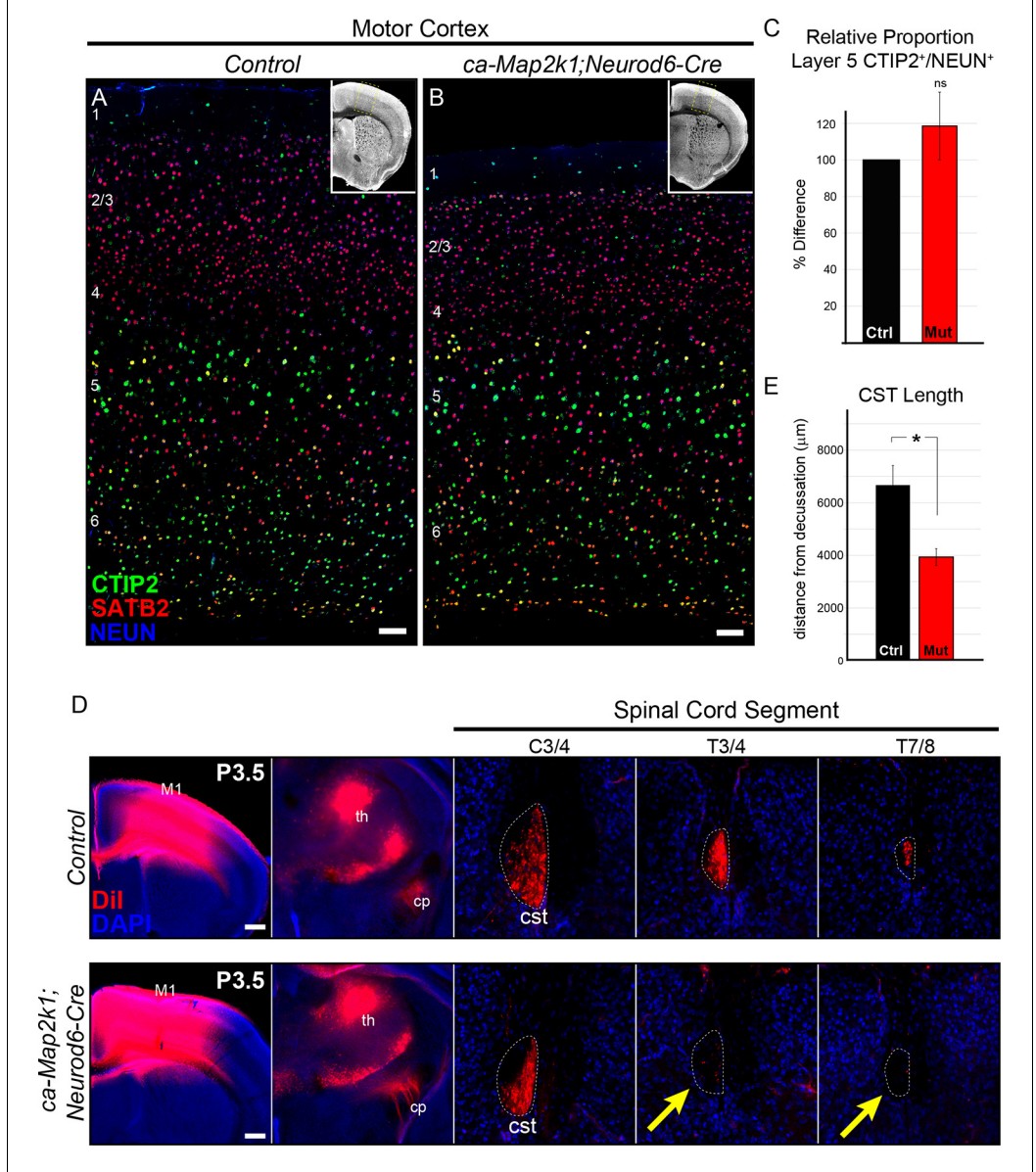

**Figure 5.** Gain of function ERK/MAPK signaling in *ca-Map2k1;Neurod6-Cre* mice decreases corticospinal extension into the spinal cord. (**A–C**) Representative confocal images of sensory cortices show that the expression and distribution of the callosal projection neuron marker, SATB2, and subcerebral projection neuron marker, CTIP2, in *ca-Map2k1;Neurod6-Cre* forebrains (**B**) appears normal when compared to littermate controls (**A**) (n=4, scale bar=100 μm). The relative proportion of CTIP2$^+$ layer 5 neurons as a percentage of NEUN$^+$ neurons within a radial unit did not show a significant difference between adult mutant and control motor cortices (**C**) (n=3, mean ± SEM, p=0.2). (**D–E**) In vivo DiI injections into P0 neonates were performed to label corticospinal axons during initial stages of elongation into the spinal cord (**D**). Analysis of DiI labeling in the spinal cord dorsal funiculus at P3 revealed a significant decrease in the extent of axonal elongation in mutant spinal cords relative to controls (**E**) (n=3, mean ± SEM, *p=0.033, scale bar=100 μm).

The following figure supplement is available for figure 5:

**Figure supplement 1.** Mouse model for excitatory neuron specific gain-of-ERK/MAPK signaling in the cortex.

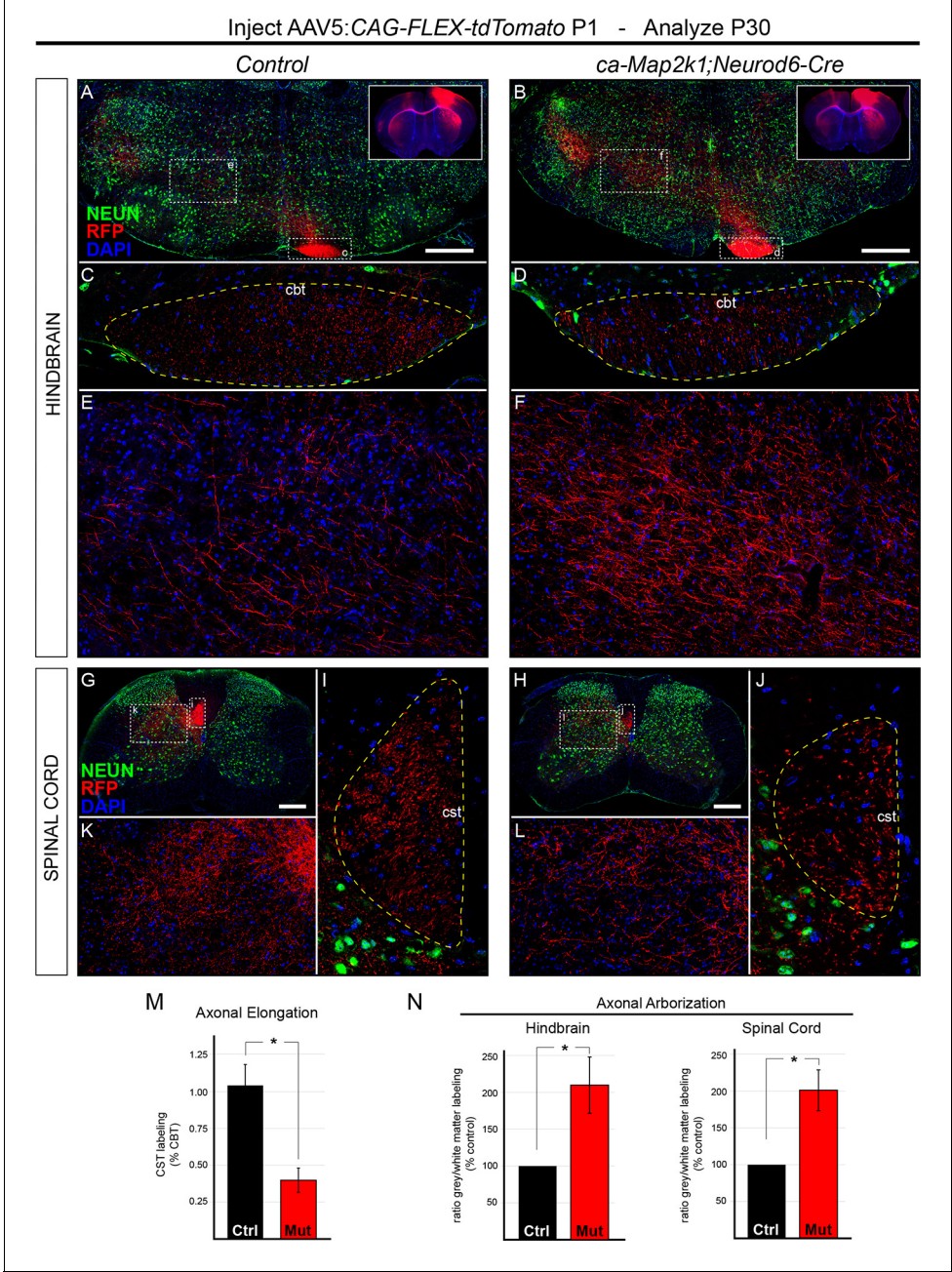

**Figure 6.** Hyperactivation of ERK/MAPK enhances axonal branching in the hindbrain and spinal cord. (A–J) AAV5-FLEX-tdTomato injections into control (*Neurod6-Cre*) and *ca-Map2k1;Neurod6-Cre* motor cortices at P1 (inset in **A–B**) results in labeling of subcerebral axon projections in P30 hindbrains (**A–F**) and spinal cords (**G–L**). To measure axonal elongation from the hindbrain to spinal cord, high resolution confocal images of the extent of axonal labeling in sections of the hindbrain corticobulbar tract (cbt, **C–D**) and brachial spinal cord corticospinal tract (cst, **I–J**) were collected and compared. The ratio of corticospinal to corticobulbar tract axonal labeling was significantly decreased in *ca-Map2k1;Neurod6-Cre* mutants (**M**), providing further evidence for a reduction in corticospinal axon elongation at P30 (n=3, mean ± SEM, *p<0.01, scale bar=500 µm). The extent of axonal branching was measured by comparing the amount of axonal labeling in the hindbrain (**A–B**, zoom in **E–F**) or spinal cord grey matter (**G–H**, zoom in **K–L**), to the amount of axonal labeling in the corresponding white matter tract, the CBT (**C–D**) or CST (**I–J**), respectively. We observed a significant increase in the relative ratio of grey/white matter labeling in *ca-Map2k1;Neurod6-Cre* mutant hindbrains (n=4, mean ± SEM, *p<0.05) and spinal cords (n=3, mean ± SEM, *p<0.05) when compared to controls (**N**).

The following figure supplement is available for figure 6:

**Figure supplement 1.** *ca-Map2k1;Emx1-Cre* mice exhibit alterations in the pattern of corticospinal outgrowth in the spinal cord similar to *ca-Map2k1; Neurod6-Cre* mutants.

relative to the hindbrain, *ca-Map2k1;Neurod6-Cre* mutant mice exhibited a significant 60.14 ± 8.3% reduction in labeling in the spinal cord CST relative to the hindbrain (*Figure 6M*). These findings provide further support for a substantial and persistent decrease in the elongation of corticospinal axons following hyperactivation of ERK/MAPK.

In striking contrast to the decreased axonal elongation into the spinal cord, we noted a significant increase in the density of axonal labeling in the hindbrain grey matter of *ca-Map2k1;Neurod6-Cre* mutants (*Figure 6A–B,E–F*). The hindbrain reticular nucleus is known to receive input from the primary motor cortex (*Esposito et al., 2014*). The ratio of axonal labeling in the hindbrain grey matter relative to labeling in the hindbrain white matter (CBT) provided quantitative evidence of enhanced axonal arborization in the *ca-Map2k1;Neurod6-Cre* mutants (*Figure 6N*). We also tested whether increased arborization could be detected in the spinal cord. The absolute level of axonal labeling density in the spinal cord grey matter was reduced in mutant mice (*Figure 6I-L*). However, a comparison of the ratio of axonal labeling in the spinal cord grey matter relative to axonal labeling in the spinal cord white matter tract suggests that CST axon arborization is increased per axon (*Figure 6N*). A similar result was also observed in the spinal cord of *ca-Map2k1;Emx1-Cre;Ai3* mutants (*Figure 6—figure supplement 1A–B,D*). In sum, our findings show that ERK/MAPK hyperactivation reduces the number of axons that extend longitudinally down the spinal cord, but the extent of arborizing axonal outgrowth into the hindbrain and spinal cord grey matter is enhanced.

## ERK/MAPK signaling is dispensable for the morphological development of cortical layer 2/3 callosal projection neurons

To definitively evaluate the requirement for ERK/MAPK signaling during callosal projection neuron development, we tested the effect of *Map2k1/2* deletion on cortical layer 2/3 neurons. Excitatory neurons in layer 2/3 primarily project to intracortical targets ipsilaterally and contralaterally through the corpus callosum. We first asked whether ERK/MAPK signaling regulates upper layer neuron number and differentiation by assessing the expression of CUX1 and SATB2, two critical transcriptional regulators of callosal projection neuron development, in P14 *Map2k1/2;Neurod6-Cre* cortices (*Alcamo et al., 2008*; *Britanova et al., 2008*). We detected a modest, but significant increase in the proportion of total NEUN⁺ neurons that express CUX1 in layer 2–4 in *Map2k1/2;Neurod6-Cre* sensory cortices (*Figure 7A–B, E*). An increase in the proportion of CUX1 labeled neurons in mutant cortices might be an expected ratiometric result of the loss of layer 5 neurons. However, other mechanisms might contribute, such as effects on intermediate progenitors (*Li et al., 2012*). SATB2 expression in upper cortical layers showed no major changes at either P1 (*Figure 3A–B*) or P14 (*Figure 1A–B*; *Figure 7C–D*). Lastly, our microarray analysis did not show decreases in a number of genes known to be expressed in upper layer cortical neurons, in fact some genes were modestly increased (*Figure 2—source data 1*). We conclude that ERK/MAPK signaling is dispensable for the survival and specification of upper layer callosal projection neurons during early excitatory circuit development.

We tested whether ERK/MAPK signaling is required for the morphological differentiation of upper layer neurons. Plasmids expressing *Neurod1-Cre* and a Cre-dependent EGFP construct were injected into the lateral ventricle of E14.5 embryos and unilaterally electroporated into developing radial progenitors. *In utero* electroporation (IUEP) of *Map2k1^{loxp/loxp};Map2k2^{-/-}* embryos at this stage of development resulted in concurrent layer 2/3 neuron-autonomous deletion of *Map2k1/2* and fluorescent labeling of a fraction of layer 2/3 dendritic and axonal arbors (*Figure 7F–I*, *Figure 7—figure supplement 1B*). This approach resulted in clear loss of MAP2K1 immunolabeling in P14 *Map2k1^{loxp/loxp};Map2k2^{-/-}* layer 2/3 neurons that had been electroporated with *Neurod1-Cre* and Cre-dependent EGFP when compared to surrounding non-electroporated neurons (*Figure 7—figure supplement 1A*). The somal size of labeled neurons was significantly decreased in *Map2k1/2* deleted layer 2/3 neurons that were electroporated with *Neurod1-Cre*, providing further evidence that ERK/MAPK signaling acts as a direct regulator of neuron cell body size (*Figure 7L*). Morphological reconstruction of dendritic arbors revealed a modest statistically significant reduction in the length of apical, but not basal, dendrites following layer 2/3-autonomous loss of ERK/MAPK signaling when compared to controls (*Figure 7F, H, K*; *Figure 7—figure supplement 1B*). Analyses of biocytin filled layer 2/3 neurons in P14-17 *Map2k1/2;Neurod6-Cre* visual cortices also revealed a modest but significant decrease in dendritic outgrowth without significant alterations in dendritic branching (*Figure 8—figure supplement 1D*).

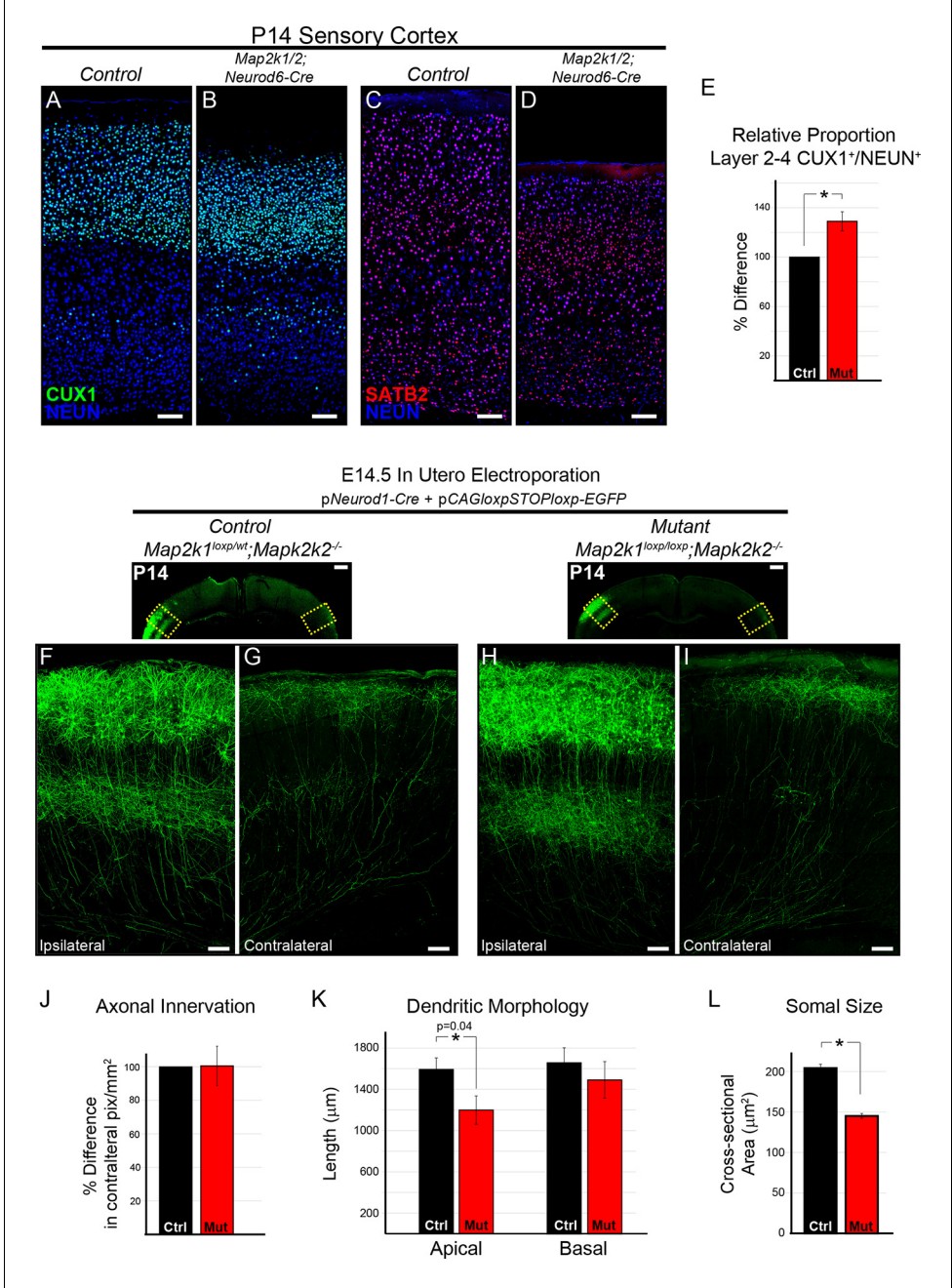

**Figure 7.** The differentiation and morphology of callosal projecting neurons in layer 2/3 does not require ERK/MAPK signaling during development. (A–E) Analysis of intracortical neuron markers, CUX1 (A–B) and SATB2 (C–D), from confocal images of P14 control (A, C) and *Map2k1/2;Neurod6-Cre* (B, D) sensory cortices did not reveal significant differences in overall expression. (E) Quantitation of the proportion of CUX1 labeled neurons in layer 2–4 as a percentage of NEUN labeled neurons in a radial unit was not decreased in mutant mice when compared to controls (n=3, mean ± SEM, *p=0.033, scale bar = 100 μm). (F–L) Cell autonomous deletion of *Map2k1/2* and labeling of a subset of layer 2/3 cortical neurons through unilateral in utero electroporation (IUEP) of p*Neurod1-Cre* and p*loxp-STOP-loxpEGFP* into the E14.5 ventricular zone. Following IUEP, coronal forebrain sections from electroporated P14 *Map2k1^{wt/loxp};Map2k2^{-/-}* control (F–G) and *Map2k1^{loxp/loxp};Map2k2^{-/-}* mutant (H–I) mice were immunostained for EGFP. Representative two-dimensional projections of confocal Z-stack images of EGFP-labeled layer 2/3 neuron bodies and dendrites in the electroporated hemisphere (F, H) and associated callosally projecting axons in the contralateral hemisphere (G, I) are shown (scale bar=50 μm). (J) The extent of axonal innervation was assessed by quantifying the number of labeled pixels within contralateral radial units. A significant difference in axonal innervation was not detected following loss of ERK/MAPK signaling (n=3, mean ± SEM, p=0.97). K. High resolution confocal Z-stacks of EGFP expressing layer 2/3 neurons were collected from control and mutant sections. Reconstruction and analysis of randomly selected layer 2/3 neurons show that the length of apical dendritic arbors was modestly reduced in *Map2k1/2* deleted layer 2/3 neurons (K) (n=10 control and 11 mutant neurons, mean ± SEM, p=0.04). No significant effect on basal dendrite length was detected in mutant neurons (mean ± SEM, p=0.625). (L) Somal size assessment revealed a substantial

*Figure 7 continued on next page*

*Figure 7 continued*
decrease in the cross sectional size of EGFP labeled layer 2/3 neuron cell bodies following IUEP mediated *Map2k1/2* deletion (n>100 neurons per condition, mean ± SEM, *p<0.001).
The following figure supplement is available for figure 7:

**Figure supplement 1.** ERK/MAPK signaling is dispensable for the differentiation and axonal morphology of callosal projection neurons in layer 2/3.

IUEP resulted in clear labeling of callosal axons in contralateral, homotypic regions of sensory cortex (*Figure 7G, I*). In contrast to corticospinal axons, no significant effect of *Map2k1/2* deletion on layer 2/3 callosal axon innervation was detected (*Figure 7G, I, J*). In a complementary approach, electroporation of *dsRed2*-expressing plasmids into E14.5 *Map2k1/2;Neurod6-Cre* mice also revealed no significant difference in the innervation of the contralateral cortex when compared to controls (*Figure 7—figure supplement 1C–H*). These data demonstrate that in callosally projecting layer 2/3 neurons, ERK/MAPK signaling cell-autonomously regulates somal size and subtle aspects of dendrite outgrowth, but not axonal morphology.

## ERK/MAPK is necessary for the expression of plasticity-associated genes and reducing intrinsic excitability during development

Many studies have shown that ERK/MAPK signaling regulates neuronal activity and synaptic plasticity (*Di Cristo et al., 2001*; *Kushner et al., 2005*; *Thomas and Huganir, 2004*). Our data show that ERK/MAPK signaling played a limited role in the morphological development of callosal layer 2/3 neurons, therefore, we tested whether the electrophysiological development of these neurons was altered in *Map2k1/2;Neurod6-Cre* mice. Indeed, gene expression profiling of P14 *Map2k1/2;Neurod6-Cre* whole cortical samples revealed diminished levels of many well-established plasticity-associated immediate early genes (IEGs), including *Arc, Fos, Npas2,4,* and *Egr1,2,4* (*Figure 8A*) (*West and Greenberg, 2011*). Western blot and immunohistochemical screening confirmed a significant decrease in ARC protein expression in mutant cortices in essentially all cortical lamina when compared to littermate controls (*Figure 8B–D*). In our P14 gene expression profiling, we also observed significantly decreased expression of many ion channels and neurotransmitter receptors that regulate neuronal excitability, including GABA/glycine receptor subunits (*Gabr-a5,-b1,-a1,-b2, Glra2*), sodium channels (*Scn3b, Scn2a1*), potassium channels (*Kcn-g1,-v1,-g3), Hcn1, Accn1, Cacna2d3,* and *Gria3*. Perhaps surprisingly, gene expression profiling of P21 gain of function *ca-Map2k1;Neurod6-Cre* cortices showed fewer significantly altered transcripts and did not reveal significant changes in plasticity- or activity-associated genes when compared to the loss-of-function cortices (*Figure 8A*). This finding supports past studies showing modest effects of MAP2K1/2 hyperactivation on neuronal and glial transcript levels (*Nateri et al., 2007*; *Sheean et al., 2014*). Possible explanations are that negative feedback loops associated with the cascade and cortical network homeostatic effects reduce the consequences of gain of function.

The observed differences in activity-dependent IEGs and ion channels indicated that electrophysiological parameters might be disrupted specifically in loss-of-function mutants. We directly tested whether ERK/MAPK was required for the normal development of intrinsic excitability using whole-cell recordings from P14 *Map2k1/2;Neurod6-Cre* sensory cortices. We observed a significant decrease in the average membrane capacitance and the membrane time constant, tau, of mutant P14 pyramidal neurons (*Figure 8—figure supplement 1A*). This result is an expected outcome from the smaller somal size of neurons in *Map2k1/2;Neurod6-Cre* cortices. No significant difference in the resting membrane potential or action potential amplitude/width was observed in mutant neurons (*Figure 8—figure supplement 1A*). However, layer 2/3 neurons fired significantly more action potentials in response to increasing amounts of injected current, demonstrating that loss of *Map2k1/2* increases layer 2/3 neuronal excitability. In mutant neurons, the action potential threshold was significantly reduced and the slope of the action potential output in response to current injection was increased by 42.6% relative to controls (*Figure 8E*, *Figure 8—figure supplement 1A*). Interestingly, a similar increase in intrinsic hyperexcitability was noted when we recorded from the surviving, presumably callosal, layer 5 neurons in P14 *Map2k1/2;Neurod6-Cre* sensory cortices (*Figure 8F*, *Figure 8—figure supplement 1A*). These data show that ERK/MAPK signaling normally

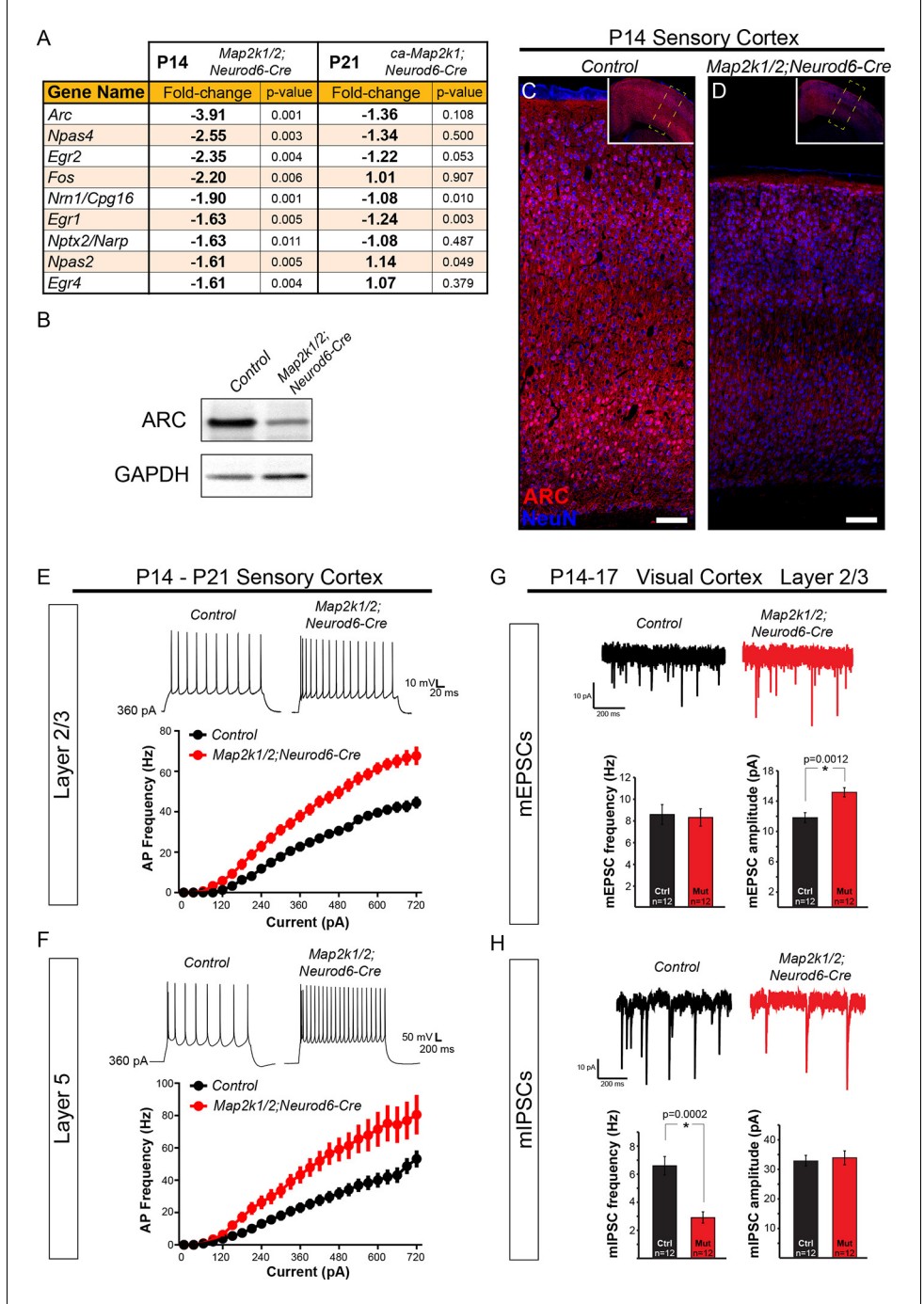

**Figure 8.** ERK/MAPK signaling promotes ARC expression and reduces neuron excitability in layer 2/3 and 5 pyramidal neurons during development. (**A**) Microarray profiling of whole cortical lysates detected a significant decrease in the expression of activity-dependent genes in P14 *Map2k1/2;Neurod6-Cre* mice compared to controls (n=3). (**B**) Western blotting of P14 control and *Map2k1/2;Neurod6-Cre* whole cortical lysates confirmed the decrease in ARC expression in mutant cortices (**B**) (n=3). (**C-D**) Immunostaining with an ARC antibody shows reduced expression across all cortical layers in P14 *Map2k1/2;Neurod6-Cre* mutants (**D**) when compared to controls (**C**) (n=3, scale bar=100 μm). (**E–F**) Average action potential frequency-current (F/I) curves recorded from layer 2/3 (**E**) and layer 5 (**F**) pyramidal neurons in acute slices of control and *Map2k1/2;Neurod6-Cre* sensory cortices. Recordings were performed in the presence of DNQX, APV, and picrotoxin to block all synaptic activity. Both layer 5 (n=16 control, 14 mutant neurons, mean ± SEM; F/I Slope t-test, p<0.0001) and layer 2/3 (n=14 control, 14 mutant neurons, mean ± SEM; F/I Slope t-test, p<0.0001) pyramidal neurons in *Map2k1/2;Neurod6-Cre* sensory cortices exhibited a marked increase in action potential firing frequency in response to increasing current injections. (**G–H**) Recordings of miniature excitatory postsynaptic currents in layer 2/3 pyramidal neurons from the visual cortex revealed a significant increase in the amplitude, but not frequency, of mEPSCs in neurons lacking *Map2k1/2* (n=12 neurons of each genotype from three independent litters, mean ± SEM, t-

*Figure 8 continued on next page*

*Figure 8 continued*

test, p=0.0012) (G). In contrast, a substantial decrease in the frequency, but not amplitude, of mIPSCs from *Map2k1/2;Neurod6-Cre* neurons was observed (H) (n=12 neurons of each genotype, mean ± SEM, t-test, p=0.0002).
The following figure supplement is available for figure 8:

**Figure supplement 1.** Loss of ERK/MAPK signaling leads to imbalanced excitatory and inhibitory synaptic drive in layer 2/3 neurons.

reduces the intrinsic excitability of both cortical layer 2/3 and layer 5 neurons during early postnatal development.

We sought to further understand how ERK/MAPK signaling regulates cortical excitability by measuring excitatory and inhibitory neurotransmission in neurons lacking *Map2k1/2.* To further minimize indirect effects due to overt loss of layer 5 neurons in sensorimotor cortices, we examined basal synaptic properties in visual cortex layer 2/3 neurons from *Map2k1/2;Neurod6-Cre* mice, where neuron number is relatively intact. We tested whether loss of ERK/MAPK signaling influences synaptic transmission by recording miniature excitatory and inhibitory postsynaptic currents (mEPSCs, mIPSCs) from layer 2/3 neurons in control and *Map2k1/2;Neurod6-Cre* cortices. Interestingly, visual cortex layer 2/3 neurons exhibited a significant increase in mEPSC amplitude, while mEPSC frequency was unaffected (*Figure 8G*). These findings suggest that loss of ERK/MAPK signaling increases the strength of glutamatergic synapses, likely through increases in postsynaptic AMPA receptor number or function. We next tested whether GABAergic neurotransmission was disrupted in *Map2k1/2;Neurod6-Cre* cortices. In contrast to spontaneous excitatory transmission, mutant layer 2/3 pyramidal neurons had a substantial decrease in mIPSC frequency, but not mIPSC amplitude (*Figure 8H*). Overall, these findings demonstrate that loss of *Map2k1/2* significantly disrupts the ratio of excitation to inhibition through increases in glutamatergic neurotransmission and decreases in GABAergic neurotransmission.

## Discussion

The pathogenesis of neurological deficits in neurodevelopmental syndromes associated with altered ERK/MAPK signaling is poorly understood, due to the lack of knowledge regarding cell-type specific effects of loss and gain of function through the ERK/MAPK pathway. Here, we have identified a striking dependence on levels of ERK/MAPK signaling for the development of large CTIP2[+] neurons in layer 5. Both high and low levels of ERK/MAPK signaling disrupted corticospinal axon projections. Interestingly, ERK/MAPK signaling does not appear to be a key regulator of projections for all excitatory neuron subtypes as axons of callosally projecting layer 2/3 neurons extend and target normally in the absence of ERK/MAPK signaling. In contrast to the neuronal subtype-specific effects on axonal outgrowth, the expression of plasticity associated proteins and neuronal intrinsic excitability were dependent on ERK/MAPK signaling in multiple cortical layers.

### Specific regulation of layer 5 CST neuron morphology

In vitro analyses have suggested that ERK/MAPK signaling acts as an important regulator of cortical neuron dendritic and axonal morphogenesis (*Dijkhuizen and Ghosh, 2005*; *Kumar et al., 2005*; *Wu et al., 2001*). However, our in vivo data reveal a remarkably specific requirement for the ERK/MAPK pathway in the morphological development of layer 5 long range projection neurons. ERK/MAPK signaling is dispensable for the early phases of CST neuronal differentiation, axon growth all the way to the medulla, and even responsiveness to midline guidance cues. However, a drastic delay in corticospinal axon growth was observed in *Map2k1/2* mutants between P2-4 when CST axons in control mice start to invade the spinal cord. By P14 in controls, CST axons have reached the lumbar spinal cord. In P14 mutant mice, only a few corticospinal axons were present in the cervical spinal cord and no axons reached lumbar segments.

Trophic cues linked to ERK/MAPK activation, specifically, BDNF, IGF1, IGF2, GDNF, and pleiotrophin, have been shown to regulate developing corticospinal neuron growth (*Dugas et al., 2008*; *Giehl et al., 1998*; *Özdinler and Macklis, 2006*; *Ueno et al., 2013*). IGF1 has been described as a particularly potent growth factor for CST development both in vitro and in vivo (*Özdinler and*

*Macklis, 2006*). IGF1 protein levels are higher in the spinal cord than the brain in the early postnatal stage (*Rotwein et al., 1988*) and we identified a focal pattern of IGF1 protein expression in the ventral medulla near the location where developing corticospinal axons enter the cervical spinal cord. However, we quite surprisingly found that deletion of *Igf1r* in long-range projection neurons did not lead to the predicted failure of CST development. Our results suggest that CST axon growth in the spinal cord may be orchestrated by multiple growth factors or via non-neuronal actions of IGF1. For example, vascular endothelial cells also express IGF1R and endothelial cell-derived trophic cues are potent regulators of corticospinal neuron survival and outgrowth in vitro (*Dugas et al., 2008*). Overall our data support the view that multiple cues converge upon ERK/MAPK to regulate transcriptional and cell biological processes required for extension of long axons.

Following the delayed entry of axons into the spinal cord in *Map2k1/2* mutants, we detected layer 5 neurons undergoing apoptosis in a restricted time frame during the first postnatal week. Counts revealed loss of roughly half of the CTIP2-expressing layer 5 neurons, which reflects a substantial proportion of the CST population. The reduced number of CST neurons in ERK/MAPK mutants is reminiscent of the loss of CST neurons after lesions to developing corticospinal projections in the neonatal period (*Merline and Kalil, 1990*; *Tolbert and Der, 1987*). Loss of trophic support from the spinal cord could plausibly account for these findings; an effect that might be exacerbated by known consequences of ERK/MAPK disruption on retrograde transport of growth factor signaling components including signaling endosomes (*Mitchell et al., 2012*). We cannot completely exclude that the reduced number of CTIP2$^+$ neurons in layer 5 results, in part, from conversion into an alternative neuronal type. For example, loss of *Fezf2* results in conversion of corticospinal neurons into a callosal fate (*Chen et al., 2005*; *Chen et al., 2005*; *Molyneaux et al., 2007*). Our assessment of SATB2$^+$/CTIP2$^-$ layer 5 neurons did not reveal a coincident increase in the number of presumed callosally fated layer 5 neurons in *Map2k1/2;Neurod6-Cre* mice. Interestingly we did not find similar layer 5 apoptosis occurring in conditional *Igf1r* mutants, placing our results at odds with a recent study suggesting that microglia-derived IGF1 is required for the survival of layer 5 long range projection neurons (*Ueno et al., 2013*). Whatever the mechanism, our data show that layer 5 corticospinal neurons are remarkably vulnerable to loss of ERK/MAPK signaling during the neonatal period and our results may be relevant to motor system dysfunction in RASopathy patients and layer 5 disorganization observed in ASDs.

Perhaps surprisingly, rostrocaudal corticospinal axon extension was also highly diminished in the setting of enhanced ERK/MAPK signaling. At every stage examined from P3 onward, the number of CST axons in the dorsal funiculus in response to gain of ERK/MAPK signaling was reduced. A distinct feature of axon growth in gain of function mutants was enhanced axonal branching in the hindbrain. Normally, over the first two postnatal weeks, ERK/MAPK activation is upregulated, coincident with increased BDNF/TrkB expression (*Maisonpierre et al., 1990*; *Timmusk et al., 1994*). BDNF/TrkB signaling has been shown to promote corticospinal axon branching in vitro (*Özdinler and Macklis, 2006*). We suggest that hyperactivation of ERK/MAPK triggers mechanisms normally associated with BDNF induced corticospinal branching, possibly resulting in premature closure of the period of rostrocaudal axon elongation. Reduced axon extension of layer 5 projection neurons in the setting ERK/MAPK hyperactivation would likely have major implications for cortical circuit development in the human brain where distances are vastly longer than in rodents.

## Global regulation of neuronal excitability

In addition to effects on axonal connectivity, dysregulated ERK/MAPK signaling is likely to disrupt glutamatergic signaling (*Di Cristo et al., 2001*; *Thomas and Huganir, 2004*). Studies have demonstrated that mature RASopathy mouse models exhibit reduced hippocampal LTP and spatial memory impairment (*Costa et al., 2002*; *Cui et al., 2008*; *Lee et al., 2014*). Our findings provide genetic confirmation for prior work using pharmacological inhibitors demonstrating ERK/MAPK regulation of plasticity-associated genes induced by excitatory activity, including *Arc, Egr2*, and *Fos* (*Majdan and Shatz, 2006*; *Panja et al., 2009*; *Tropea et al., 2006*; *Waltereit et al., 2001*). Further, we establish a link between ERK/MAPK and expression of *Npas4, Nrn1/Cpg16*, and in vivo (*Coba et al., 2008*). Unexpectedly, hyper-activation of ERK/MAPK signaling with the *ca-Map2k1$^{S217/221Q}$* mutant had little effect on the expression of these same plasticity associated genes. These findings are in line with a past study of hippocampal gene expression in a different ca-MAP2K1 mouse mutant (*Nateri et al.,*

*2007*). The severity of effects on expression of plasticity associated genes is likely correlated with the magnitude of ERK/MAPK hypo- or hyper-activity in response to specific mutations.

Since ERK/MAPK dependent changes in plasticity associated gene expression were widespread, we asked whether ERK/MAPK has general or layer-specific effects on developing projection neuron excitability and synaptic transmission. Our work demonstrates that complete loss of ERK/MAPK signaling leads to increased intrinsic excitability of both layer 2/3 and layer 5 pyramidal neurons. Further, we show that *Map2k1/2* deleted layer 2/3 neurons exhibited increased excitatory synaptic strength. We conclude that ERK/MAPK signaling contributes to excitatory/inhibitory balance during the early postnatal period by reducing the excitability of cortical excitatory neurons. Changes in excitatory drive in a subset of layer 2/3 neurons have been shown to trigger a compensatory homeostatic increase in inhibitory drive, thus maintaining a stable synaptic excitatory/inhibitory ratio (*Xue et al., 2014*). Remarkably, layer 2/3 pyramidal neurons lacking *Map2k1/2* exhibit reduced inhibitory synaptic input in the absence of a fully compensatory change in excitation. Loss of ERK/MAPK activity in cortical pyramidal neurons may disrupt the homeostatic balance between synaptic excitation and inhibition, a mechanism hypothesized to form the neurological basis of autism (*Rubenstein and Merzenich, 2003*). ERK/MAPK may also regulate the release of BDNF from excitatory neurons, a well-known regulator of inhibitory synapse formation (*Huang et al., 1999*; *Kohara et al., 2007*; *Porcher et al., 2011*).

During postnatal cortical pyramidal neuron development, input resistance decreases and the frequency of GABAergic and amplitude of glutamatergic spontaneous neurotransmission increase (*Desai et al., 2002*; *Morales et al., 2002*). Our findings are consistent with pyramidal neurons lacking *Map2k1/2* failing to undergo these developmental alterations and further suggest that loss of ERK/MAPK signaling may arrest normal physiological development. Importantly, the alterations in action potential threshold in neurons lacking *Map2k1/2* may secondarily result from increased sodium channel density due to the smaller neuronal size or the observed increase in membrane resistance, which may also influence quantal amplitude measurements. Our observations that disruption of ERK/MAPK signaling alters excitatory/inhibitory balance and the expression of select plasticity associated genes, coupled with known effects on Hebbian forms of synaptic plasticity (*Thomas and Huganir, 2004*), suggest that alterations in ERK/MAPK signaling are likely to dramatically disrupt network activity and cortical re-wiring.

## Circuit abnormalities in RASopathies

Hypotonia, muscle weakness, and delay in motor milestones are often observed in RASopathy patients (*Dileone et al., 2010*; *Mejias et al., 2011*; *Oberman et al., 2012*; *Stevenson et al., 2012*; *Tidyman et al., 2011*). Although some of these symptoms may be due to alterations in muscle development, we have shown here that upper motor neuron and corticospinal tract development are especially sensitive to both gain and loss of ERK/MAPK activity. Thus, deficits in motor function or motor learning in Ras/MAPK Syndromes may be explained, in part, by altered corticospinal connectivity. Importantly, our data point to distinct effects in gain vs loss of function mutants. Our results therefore argue for a mutation-specific approach to correct neurological dysfunction within the RASopathy spectrum. Aberrant long-range circuit development has also been proposed as a defining feature of ASD pathogenesis and that layer 5 may be a focal point (*Stoner et al., 2014*; *Willsey et al., 2013*). An interesting possibility is that altered activation of ERK/MAPK at early developmental stages in response to certain ASD linked mutations (*Fmr1*, *Mecp2*, *Tsc1/2*) and environmental insults (hypoxia, inflammation, etc.) might contribute to select cortical circuit abnormalities, especially for neurons projecting over long distances.

# Materials and methods

## Transgenic mice

Animal experiments were performed in accordance with established protocols approved by the Institutional Animal Care and Use Committee at the University of North Carolina–Chapel Hill and Arizona State University and NIH guidelines for the use and care of laboratory animals. All mice were housed in standard conditions with food and water provided ad libitum and maintained on a 12 hr dark/light cycle. Experiments were replicated a minimum of three times with mice derived from independent

litters. *Neurod6-Cre* or *Emx1-Cre* expression alone did not have a detectable effect on the phenotypes described in this manuscript. Thus, Cre-expressing or Cre-negative littermates were utilized as controls unless indicated otherwise. *Map2k1*$^{loxp/loxp}$mice possess a loxp flanked exon 3 while *Map2k2*$^{-/-}$ mice contain a neo insertion in exons 4–6, which encodes the kinase domain (*Belanger et al., 2003*; *Bissonauth et al., 2006*). *Map2k2*$^{-/-}$ mice are viable and breed normally. *Loxp-STOP-loxp-caMAP2K1* mice were kindly provided by Dr. Maike Krenz and Dr. Jeffrey Robbins (*Krenz et al., 2008*); the *Igf1r*$^{loxp/loxp}$ were kindly provided by Dr. Ping Ye (*Liu et al., 2009*); the *Neurod6-Cre* mice were kindly provided by Dr. Klaus Nave and Dr. Sandra Goebbels (*Goebbels et al., 2006*); and the *Emx1-Cre* mice were kindly provided by Dr. Franck Polleux (*Gorski et al., 2002*). *Ai3* mice were purchased from Jackson laboratories (*Madisen et al., 2010*). All mice in this study were of mixed genetic background.

Genomic DNA extracted from tail or toe samples was utilized for mouse genotyping by PCR using standard techniques. Primers for gene amplification are as follows (listed 5'-3'): Cre - TTCGCA-AGAACCTGATGGAC and CATTGCTGTCACTTGGTCGT amplify a 266 bp Cre allele; *Map2k1* –CA-GAAGTTCCCACGACACTA, CTGAAGAGGAGTTTACGTCC, and GTCTGTCACTTGTCTTCTGG amplifies a 372 bp wild type and a 682 bp floxed allele; *Map2k2* – CTGACCTTCCTGTAGGTG, ACT-CACGGACATGTAGGA, and AGTCATAGCCGAATAGCCTC amplify a 293 bp wild-type allele and a 450 bp knockout allele; caMAP2K1 -GTACCAGCTCGGCGGAGACCAA and TTGATCACAGCAATG-CTAACTTTC amplify a 600 bp mutant allele; *Ai3* – AAGGGAGCTGCAGTGGAGTA, CCGAAAATCT-GTGGGAAGTC, ACATGGTCCTGCTGGAGTTC, and GGCATTAAAGCAGCGTATCC amplify a 297 bp wild-type allele and a 212 bp Ai3 allele; *Igf1r*-CTTCCCAGCTTGCTACTCTAGG and CAGGCTTG-CAATGAGACATGGG amply a 124 and a 220 bp band for wild-type and floxed alleles.

## Viral and DiI injections

P0-P5 litters were removed as a group, cryo-anesthetized on wet-ice for 3–5 min, and immediately injected with 50-500nl of solution using a 5 uL Hamilton syringe fitted with a 32 gauge beveled needle mounted to a stereotaxic arm. For viral labeling, the AAV5-CAG-FLEX-tdTomato vector was prepared by the UNC Viral Vector Core and diluted in sterile PBS, 5% sorbitol, and 0.1% Fast Green to allow for visualization prior to injection. For DiI tracing, a 10% DiI solution (Life Technologies) was prepared in DMSO and injected into the primary motor cortex or the cervical spinal cord. Upon completion of the injection, pups recovered on a heating pad and were returned as a group to the home cage.

## In utero electroporation

In utero electroporations were performed as previously described (*Li et al., 2012*) with a few modifications. In brief, E14.5 timed-pregnant females were anesthetized with isoflurane and the uterine horn was accessed through a cesarean section procedure. The lateral ventricles of the embryos were injected with 1–2 µg of plasmid prepared using an EndoFree plasmid purification kit (Qiagen) diluted in 1 x PBS with 0.1% Fast Green dye for visualization. Five electrical pulses were delivered at 30V (50 ms duration) with a 950 ms interval using 5 mm paddle electrodes. The uterine horns were then gently reinserted into the abdominal cavity and the abdomen wall and skin was sutured. Electroporated mice were sacrificed at the appropriate time point and processed for analysis.

## Tissue preparation

Mice of the appropriate age were anesthetized and perfused transcardially with 4% paraformaldehyde/PBS. For cryoprotection, subdissected samples were incubated in a graded series of 10%, 20%, and 30% sucrose/PBS at 4°C before embedding in O.C.T. compound and freezing. Cryostat sections were collected on Fisherbrand Superfrost/Plus slides (Fisher Scientific) and air-dried prior to staining. For some experiments, brains were dissected, postfixed, and mounted in agarose prior to vibratome sectioning.

## Immunolabeling

For immunofluorescent staining, sections were rinsed in PBS and blocked with 5% normal serum/0.1% Triton X-100/PBS at room temperature. Primary antibodies were diluted in blocking solution and incubated 1–2 days at 4°C with gentle agitation. The antibodies utilized were; rabbit anti-

Parvalbumin (Swant), chicken anti-EGFP (Aves Labs), rat anti-CTIP2 (Abcam), rabbit anti-SATB2 (Abcam), rabbit anti-CUX1 (Santa Cruz), mouse anti-NEUN (Chemicon), rabbit anti-Cleaved Caspase-3 (Cell Signaling Technology), rabbit anti-IBA1 (Wako), goat anti-IGF1 (R&D Research), rabbit anti-PKCγ (Santa Cruz), rabbit anti-MAP2K1/2(MEK1/2) (Abcam), rabbit anti-P-MAPK1/3(ERK1/2) (Cell Signaling Technology) and rabbit anti-IGF1Rβ (Cell signaling Technology). After rinsing in PBS/T, the secondary antibody was diluted in blocking solution and added overnight at 4°C. Secondary antibodies included Alexa Fluor 488, 546 or 568, and 647 conjugated anti-rabbit, anti-mouse, anti-rat, or anti-goat IgG (Invitrogen). For some experiments slides were then incubated in Hoechst or DAPI for nuclear labeling, rinsed, and mounted. Images were collected with a Zeiss LSM 710, 780, or Leica SP5 laser scanning confocal microscope.

## Image analysis and quantitation

Confocal images of regions of interest were collected from individual brain sections for each animal. For assessment of relative neocortical volume, cortical area was measured in five anatomically matched coronal sections along the rosto-caudal axis, averaged, and normalized against the control brain. For assessment of CTIP2, CUX1, SATB2, and NEUN expressing cells, regions of primary sensory, motor, and visual cortex from at least three anatomically matched sections of mutant and control cortices were defined using morphological and anatomical features. Radial columns were outlined within the cortical region of interest and measured. The total sampled area from a specific region of cortex of a single brain ranged from 1–3 mm$^2$. Individual layer boundaries were determined by the changes in density and appearance of NEUN labeling. Images were then transferred into ImageJ, a pixel intensity threshold was set manually by an observer blind to the genotype, and watershed segmentation was performed. The binary image mask was then analyzed using the particle analysis tool to count the number of events with a min-max size cutoff of 50–600 µm$^2$ for NEUN labeling and 30–300 µm$^2$ for CTIP2$^+$, CUX1$^+$, or SATB2$^+$ labeled nuclei. For NEUN density determination, the total number of NEUN$^+$ cells within a radial column was divided by the area of the column and averaged across at least three separate columns within the cortical region of interest. For determination of the relative proportion of CTIP2$^+$ or SATB2$^+$ cells in layer 5 or CUX1$^+$ cells in layers 2–4, the number of each labeled cell within specific laminar boundaries was divided by the total NEUN count within the entire radial column to determine the proportion of each neuronal subtype per radial column. Results from this analysis were averaged across three individual radial columns per cortical region and normalized against the littermate control analyzed in parallel for each mutant. These images were also utilized for determination of the cross sectional area of neuronal soma. Randomly selected, well-labeled NEUN expressing neurons that included a DAPI labeled nucleus were outlined manually in Photoshop and measured.

For analysis of axonal innervation, a modification of the cell counting procedure described above was utilized where images of the region of axonal innervation from at least three anatomically matched sections were collected, manually thresholded in ImageJ, and the number of labeled pixels was measured. For measurement of *ca-Map2k1;Emx1-Cre* spinal cords background signal was determined by measuring a non-innervated area in the same section and subtracted from the mean intensity of an anatomical region of interest.

Representative images have been cropped and adjusted for brightness and contrast in Photoshop for presentation. Student's t-test was used for statistical analysis.

## Western blotting

Sensorimotor cortices were dissected from both mutant and litter mate control mice and lysed inRIPA buffer (0.05M Tris-HCl, pH 7.4, 0.5M NaCl, 0.25% deoxycholic acid, 1% NP-40, and 1 mM EDTA, Millipore) supplemented with 0.1% SDS, protease inhibitor cocktail (Sigma) and phosphatase inhibitor cocktail II and III (Sigma). Lysates werecleared by centrifugation and protein concentration was determinedusing the Bio-Rad protein assay (Bio-Rad) using BSA as a standard. Equal amounts of protein were denatured in reducing sample buffer, separated by SDS-PAGE gels,and blotted to PVDF membranes (Bio-Rad). Blots were blocked with 5% BSA in TBS containing 0.5% Tween 20 (TBS-T) for 1 hr at roomtemperature, then incubated overnight at 4°C with primaryantibodies. The primary antibodies used were rabbit anti-phospho MAPK1/3(ERK1/2) (Thr202/Tyr204) (Cell Signaling Technology, Inc), rabbit anti-MAPK1/3(ERK1/2) (CST), rabbit anti-phospho-p90RSK (Thr573) (Cell

Signaling Technology, Inc.), rabbit anti-RSK (Cell Signaling Technology, Inc.), rabbit anti-MSK1 (Ser360) (Abcam), rabbit anti-MSK1 (Cell Signaling Technology, Inc.), rabbit anti-MAP2K1/2(MEK1/2) (Cell Signaling Technology, Inc.), rabbit anti-ARC (Synaptic System) and anti-GAPDH (Cell Signaling Technology, Inc.). After washing with TBS-T, membraneswere incubated with HRP-conjugated secondary antibodies in 5% milk in TBS-T for 2 hr at room temperature. Blots were washed with TBS-T and detection wasperformed with SuperSignal West Pico chemiluminescent substrate (Thermo Scientific).

## Microarray profiling

Dorsal cortices from control and mutant embryos derived from two independent litters at P9 and three independent litters at P14 were dissected and total RNA was extracted with Trizol (Invitrogen), followed by mRNA extraction with a Qiagen RNeasy Mini kit per manufacturer's instructions. RNA was assayed for quality and quantity with an Agilent 2100 Bioanalyzer and a Nanodrop spectrophotometer. Total RNA was amplified, labeled, and hybridized on Affymetrix arrays in the UNC Functional Genomics Core. Slides were processed and scanned according to the manufacturer protocol. Data was further processed using the RMA algorithm for background adjustment, quantile normalization, and median polish probeset summarization. P14 and P21 cortical samples were $log_2$ transformed and differential expression and p-values were calculated using a mixed-model ANOVA in Partek Genomics Suite. Transcripts demonstrating an average change in expression that is >1.5 fold (up-regulated) or < −1.5 fold (down-regulated) and a p-value < 0.05 were considered differentially expressed. To examine *Map2k1* exon 3 expression specifically, individual probe ID#s 561826 (TGGTTCCGGATTGCGGGTTTGATCT),1056599 (TCCGGATTGCGGGTTTGATCTCCAG), and 1079432 (CCGGATTGCGGGTTTGATCTCCAGG) were processed as described without probeset summarization. All microarray data from this study have been deposited into the NCBI GEO Database under accession number GSE75129.

## Electrophysiology

For cortical slice preparation, mice were anesthetized with pentobarbital sodium and decapitated after disappearance of corneal reflexes. Brains were rapidly removed and 350 µm coronal sections were cut using a vibrating microtome (Leica VT1200S) in ice-cold dissection buffer containing (in mM): 87 NaCl, 2.5 KCl, 1.25 NaH$_2$PO$_4$, 25 NaHCO$_3$, 75 sucrose, 10 D-(+)-glucose, 1.3 ascorbic acid, 7 MgCl$_2$ and 0.5 CaCl$_2$, bubbled with 95% O$_2$ and 5% CO$_2$. Following dissection, slices recovered for 20 min at 35°C in artificial cerebrospinal fluid (ACSF) containing (in mM): 124 NaCl, 3 KCl, 1.25 Na$_2$PO$_4$, 26 NaHCO$_3$, 1 MgCl$_2$, 2 CaCl$_2$ and 20 D-(+)-glucose, saturated with 95% O$_2$, 5% CO$_2$ and were then kept at 21–24°C for at least 40 min. During this period and before transferring sections to the recording chamber, slices were maintained in ACSF supplemented with 1.25 mM ascorbic acid. Recordings were made in a submersion chamber at 32°C +/- 1°C in ACSF, except where noted.

Patch pipettes were pulled from thick-walled borosilicate glass with open tip resistances of 2–7 MΩ. Neurons were recorded in either voltage- or current-clamp configuration with a patch clamp amplifier (Multiclamp 700A), and data were acquired and analyzed using pCLAMP 10 software (Molecular Devices). L2/3 or L5 neurons were visually identified with infrared differential interference contrast optics. All recordings were performed from excitatory neurons as verified by the presence of dendritic spines and/or an apical dendrite oriented towards the pial surface. To minimize variability, recordings were performed from a mutant and control littermate with identical, same day conditions.

## Voltage-clamp recordings

Voltage-clamp recordings were performed from visual cortical layer 2/3 neurons, as described previously (*Larsen et al., 2011*). Events with a rapid rise time and exponential decay were identified as mEPSCs or mIPSCS using an automatic detection template in pCLAMP 10. To isolate mEPSCs, slices were placed in a submersion chamber, maintained at 32°C and perfused at 2 mL/min with oxygenated ACSF containing 200 nM tetrodotoxin and 50 µM picrotoxin. Pipettes were filled with internal solution containing (in mM): 100 K-Gluconate, 20 KCl, 0.2 EGTA, 4 Mg-ATP, 0.3 Na-GTP, 10 HEPES, 10 Na-phosphocreatine, and 0.5% (w/v) neurobiotin with pH adjusted to 7.25 with 1M KOH and osmolarity adjusted to ~300 mOsm by addition of sucrose. To isolate mIPSCs, ACSF included 20 µM

6,7-dinitroquinoxaline-2,3-dione (DNQX), 100 µM D,L-2-amino-5-phosphonopentanoic acid (D,L-AP5), and 200 nM tetrodotoxin. Pipettes were filled with a high internal chloride concentration to increase the chloride driving force at -70 mV and contained (in mM): 134 KCl, 2 NaCl, 10 HEPES, 0.2 EGTA, 10 sucrose, 4 Mg-ATP, 0.3 Na-GTP, 14 Naphosphocreatine, 0.5% (w/v) neurobiotin, with pH adjusted to 7.2 with KOH and osmolarity adjusted to ~300 mOsm by addition of sucrose. All voltage-clamp recordings were sampled at 10 kHz and Bessel filtered at 2 kHz. Series and input resistances were monitored throughout the experiments by measuring the response to a −5 mV step during each sweep. Series resistance was calculated using the capacitive transient at the onset of the step and input resistance was calculated from the steady-state current during the step. No series resistance compensation was applied.

## Current-clamp recordings

Intrinsic excitability recordings were performed in L2/3 and L5 somatosensory cortex with AMPA/Kainate, NMDA, and GABA(A) receptor-mediated synaptic transmission blocked by adding 20 µM DNQX, 100 µM D,L-AP5, and 50 µM picrotoxin to the ACSF. Throughout the experiment, current was injected to maintain a −70 mV resting potential. Action potential threshold was defined as the voltage at which dv/dt=20 V/s. Adaptation ratio was measured as the ratio of the third interspike interval to the last interspike interval at currents which evoked 6–7 action potentials to minimize confounds in adaptation ratio due to differences in action potential number between genotypes. Action potential amplitude was measured relative to the membrane potential prior to current injection (-70 mV). Action potential slope was analyzed by comparing threshold currents for generating action potentials to responses at two times the threshold current, which was divided by the difference in current levels. Input resistance was calculated by measuring the steady-state response to a 50 pA, 300 ms step at each current injection and was averaged across all current injections. Resting membrane potential was measured at the onset of the recording. Current clamp recordings were sampled at 20 kHz and Bessel filtered at 10 kHz. Membrane capacitance and time constant values (*Figure 8—figure supplement 1*) were calculated in voltage-clamp at the onset of the recording. Capacitance values were measured from the averaged integrated transient capacitive current produced during a 10 mV step sampled at 100 kHz in membrane test mode. Membrane time constant values were measured from the transient portion of the current response to the voltage step and were fit by a single exponential.

## Acknowledgements

We are grateful to E. Anton, F. Polleux, and the members of the Snider and Philpot labs for many helpful discussions and providing transgenic mice. We thank Vladimir Ghukasyan, Colin Parker, Becca Reinking-Herd, Sam Lusk, Anna Krueger, Julia Pringle, Meghan Morgan-Smith, and Cyril Justin Dizon for technical assistance and Ping Ye (UNC-Chapel Hill) for generously sharing the *Igf1r* conditional knockout mice. We also thank the UNC Confocal and Multiphoton Imaging Core, Functional Genomics Core, Viral Vector Core, and Expression Localization Core Facilities funded by NINDS Center grant P30 NS045892 and U54HD079124; and the ASU WM Keck Bioimaging Facility in the School of Life Sciences funded by NIH S10-RR027154 for technical support.

## Additional information

### Funding

| Funder | Grant reference number | Author |
|---|---|---|
| National Institute of Neurological Disorders and Stroke | K99/R00-NS076661 | Jason M Newbern |
| National Institute of Neurological Disorders and Stroke | R01-NS031768 | William D Snider |

| National Institute of Neurological Disorders and Stroke | R01-NS085093 | Benjamin D Philpot |
| --- | --- | --- |
| Simons Foundation | SFARI#274426 | Benjamin D Philpot |
| Children's Tumor Foundation | #2015-01-013 | Lei Xing |

The funders had no role in study design, data collection and interpretation, or the decision to submit the work for publication.

## Author contributions

LX, XL, YW, WDS, JMN, Conception and design, Acquisition of data, Analysis and interpretation of data, Drafting or revising the article, Contributed unpublished essential data or reagents; RSL, GRB, BDP, Conception and design, Acquisition of data, Analysis and interpretation of data, Drafting or revising the article

## Author ORCIDs

Rylan S Larsen, http://orcid.org/0000-0002-7136-1175

## Ethics

Animal experimentation: This study was performed in strict accordance with the recommendations in the Guide for the Care and Use of Laboratory Animals of the National Institutes of Health. All of the animals were handled according to approved institutional animal care and use committee (IACUC) protocols of the University of North Carolina (#14-006) and Arizona State University (#14-1328). All surgeries were performed under appropriate anesthesia and every effort was made to minimize suffering.

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
