## [Decision Letter]

Thank you for submitting your work entitled "Layer-specific and general requirements for ERK/MAPK signaling in the developing neocortex" for consideration by *eLife*. Your article has been reviewed by three peer reviewers, one of whom, Sacha Nelson, is a member of our Board of Reviewing Editors. The evaluation has been overseen by a Reviewing Editor and a Senior Editor.

The reviewers have discussed the reviews with one another and the Reviewing editor has drafted this decision to help you prepare a revised submission.

Summary:

The authors investigate the role of altered RAS/MAPK signaling in neurons of the postnatal mouse cortex. The authors use genetics to study the effects of loss and gain of function of this signaling pathway in pyramidal neurons and find that both manipulations impair normal formation of the corticospinal tract. In addition, loss of function manipulations has more subtle physiological effects on other classes of pyramidal neurons. Because this signaling pathway has previously been implicated in developmental brain disorders in humans, the present results are interesting for understanding how some of the features of these disorders arise.

Essential revisions:

1) All three reviewers were concerned about the interpretation of the apparent loss of CTIP^+^ corticospinal neurons. Although there is some evidence for cell death, the number of activated caspase positive neurons is small and the authors cannot rule out the possibility that some neurons have converted from a CTIP^+^, corticospinal fate to callosally projecting fate. Although reduced *Fezf2* expression is interpreted as reflecting a loss of these neurons, it could also have been down regulated which may cause neurons to switch their fate. Although it would be preferable to gain mechanistic insight into this question by performing additional experiments, the reviewers agreed that this could probably be addressed through textual changes.

As each reviewer raised some additional points which you may or may not choose to address, I have included the full reviews below.

Reviewer #1:

The authors use mouse genetics to study the effects of loss and gain of function of ERK/MAPK kinase signaling in the neocortex and find that both manipulations impair normal formation of the corticospinal tract. In addition, loss of function manipulations has more subtle physiological effects on other classes of pyramidal neurons. Because this signaling pathway has previously been implicated in developmental brain disorders in humans, the present results are interesting for understanding how some of the features of these disorders arise.

My major concern about the paper in its present state is with respect to the interpretation of the results. The authors find that the cortex is reduced in volume but not in cell number. They then find that CTIP^+^ neurons (that normally form the pyramidal tract) are greatly reduced in motor and somatosensory cortex, but are relatively unaffected in visual cortex (where this same class of neurons does not normally persist in projecting to the spinal cord). The interpretation is that there has been a selective loss the CTIP^+^ neurons. In support of this there is an increase in activation of caspase 3, however the number of neurons with activated caspase 3 is small and the authors cannot rule out the possibility that some neurons have converted from a CTIP^+^, corticospinal fate to callosally projecting fate. I do not think that this is less interesting, or that the authors need to do additional experiments to sort out the relative contributions of cell death and change in cell fate. But they should acknowledge this likely possibility. One additional reason for thinking this likely is that it appears in Figure 2 that the vast majority of the YFP16 neurons are thick-tufted L5 neurons (i.e. the large corticofugal neurons with a tuft in layer 1) whereas in the mutant these neurons take on the morphology (and presumably axonal projections and physiology) of the other major L5 population, the callosal neurons. This transformation occurs in knockout of the transcription factor Fezf2, which is upstream of CTIP2 expression.

Reviewer #2:

In the manuscript titled, "Layer-specific and general requirements for ERK/MAPK signaling in the developing neocortex," Xing et al. investigate the role of altered RAS/MAPK signaling in neurons of the postnatal mouse cortex. This signaling pathway is altered in many neurodevelopmental disorders, collectively termed RASopathies, yet it remains unclear how increased and/or decreased activation of this pathway alters neuronal function. Through understanding this relationship, insights are made into the mechanisms underlying these neurodevelopmental disorders. Therefore, the questions addressed in this manuscript are of high importance, and may have broad interest in the general biomedical sciences.

The senior authors have an extensive track record studying the effect of altered ERK signaling using loss and gain of function genetics. In this article, the authors focus on post-mitotic postnatal cortical neurons by conditionally removing Mek1 in a Mek2 null background utilizing previously described Nex-cre mice, which do not express cre in glial cells, proliferating progenitors, nor inhibitory neurons (Goebbels et al., 2006) as well as using the EMX1-cre line which is more broadly expressed in multiple cell types of the forebrain, while excluding inhibitory neurons (Gorski et al., 2002).

The novel findings in this paper include the observation that loss of ERK/MAPK signaling in immature neurons leads to a decreased number of layer 5/CTIP2^+^ neurons in motor- and sensory-, but not visual-cortex. They further show a selective loss of layer 5 cortico-spinal tract deep projection neurons, preceded by apoptosis activation. In addition, they show that IGF1 and IGF1R are not significant participants in this phenotype. Surprisingly, they observe that gain of ERK/MAPK signaling also leads to projection neuron reductions, as well as increased arborization.

The authors also examined layer 2/3 neurons. Loss of MEK1/2 also decreased their soma size, as well as decreased dendrite length. Additional novel observations include a decrease in plasticity-associated genes, reduced AP threshold in 2/3 neurons, increased mEPSC amplitude and decreased mIPSC frequency.

The scientific methods, number of replicates, statistically analyses, and overall writing of this manuscript are of high quality. One main weakness in this paper is that all the observed phenotypes can be attributed to poor cell survival, and not other more specific mechanisms such as a decrease in plasticity genes. Also, the direct link between MEK and survival is not addressed. The following issues should be addressed before publication to clarify this manuscript:

1) The authors show that pERK is upregulated in cortex via Western blot. Is pERK selectively increased in layer 5 neurons in control postnatal mice, and as such, is that why this cell type is dramatically affected by the mutation? This should be examined with immunohistochemistry.

2) It is interesting that the layer 5 neurons in the visual cortex are not affected. Is the phenotype activity dependent? This makes sense since visual experience occurs after somatosensory experience. The manuscript would be less descriptive and improved by investigating this issue. One possibility would be to compare layer 5 in the barrel cortex from mice with and without whiskers removed.

3) Is cell death also occurring in 2/3? This should be analyzed with caspase-3 activation assay and added to the manuscript.

4) The reduction in AP threshold and increased mEPSC amplitude could arise due to smaller soma size, via increased sodium channel density and/or increased input resistance. Please mention this in the Discussion.

5) Explore why mIPSC frequency (in Visual Cortex Layer 2/3) is decreased despite the lack of cre expression in inhibitory neurons. Is this due to the deficiency of ERK/MAPK signaling postsynaptically causing decreased GABAergic synapse number? The authors should investigate this by immunostaining for GABA receptors on 2/3 neurons or inhibitory synapse number by co-localization of a pre and postsynaptic marker around 2/3 cell bodies with immunochemistry.

Reviewer #3:

This is a nice paper examining the role of ERK/MAPK pathway in neocortical development and corticospinal tract (CST) formation. They show that the number of Bcl11b/Ctip2-positive neurons in layer 5 and CST axons length decrease with both, reduced and increased expression of *Mek1/2*, a mediator of ERK/MAPK pathway. Apart from effects of ERK/MAPK pathway on CST, the authors show that *Mek1* deletion, enhanced intrinsic excitability in both layers 2/3 and 5 and created imbalanced synaptic excitation and inhibition.

The paper is well written, and the experiments are generally well performed and easy to interpret. I have a few quibbles about the experiments and think some text revisions are in order, but overall this is a nice contribution to the literature that should be well received.

Major concern:

1) What mediates ERK/MAPK pathway function in layer 5 corticospinal neurons remains unanswered. Moreover, this study contradicts earlier findings by Ozdinler and Macklis, 2006, where they show that IGF1/IGFR1 drives CST formation in vitro and in vivo. In my opinion, the authors failed to take the full advantage of their microarray dataset to address this particular issue. First, their microarray data shows that *Bcl11b/Ctip2* expression is maybe slightly decreased, which is consistent with their other data. On the other hand, the expression levels of *Fezf2* (also known as *Fezl* or *Zfp312*) (PMID: 16157277; PMID: 16314561; PMID: 16284245) and *Sox5* (PMID: 18840685; PMID: 18215621), which were previously shown to be necessary for the specification of layer 5 projection neurons and the formation of CST, are dramatically reduced. Also, inactivation of Fezf2 leads to reduced soma size and dendritic arborization of layer 5 subcerebral (corticospinal) neurons (PMID: 16314561).Thus, it is possible that the two transcription factors are involved in mediating the effects of the ERK/MAPK pathway in layer 5 corticospinal neurons. Moreover, the authors speculate in paragraph one, subheading “ERK/MAPK signaling is necessary for corticospinal axon extension”, that the effects are postmitotic as "Nex:Cre is not expressed until neurons are post-mitotic". Actually the SOX5 is expressed only postmitotically/postmigratory in layer 5 projection neurons according to PMID: 18840685, further reinforcing this possibility. Thus, the link to *Fezf2* and *Sox5* should be explored further as a possible mechanism. One possible experiment would be to overexpress or reduce *Fezf2* or *Sox5* in layer 5 neuron of *Mek1/2cKO* or *caMEK1*, respectively.

---

## [Author Response]

*Summary: The authors investigate the role of altered RAS/MAPK signaling in neurons of the postnatal mouse cortex. The authors use genetics to study the effects of loss and gain of function of this signaling pathway in pyramidal neurons and find that both manipulations impair normal formation of the corticospinal tract. In addition, loss of function manipulations has more subtle physiological effects on other classes of pyramidal neurons. Because this signaling pathway has previously been implicated in developmental brain disorders in humans, the present results are interesting for understanding how some of the features of these disorders arise.*

We thank you and the reviewers for the positive and helpful comments about our manuscript. All of the reviewers were concerned about the possible conversion of corticospinal neurons to an alternative fate in the *Map2k1/2(Mek1/2)* deleted cortices. In the revised version of the manuscript, we now provide additional analyses and textual revisions to address this point as outlined below under Essential revisions. Moreover, we have provided detailed responses to all of the reviewer’s individual concerns, and in some cases added additional data.

In the course of revising the manuscript we discovered a minor error in the presentation of the gene expression profiling data in Figure 2—figure supplement 1 of the original manuscript. These data were originally described as P14 when they are actually from P9 cortices. Our P14 data set was inadvertently omitted, as were p-values. In the revised manuscript, we now include results from both the P9 and the P14 gene expression profiling together with p-values in Figure 2—figure supplement 2. The raw microarray data has been submitted to the GEO database and the accession number (GSE75129) has been added to the revised manuscript. Both the P9 and P14 results indicate that numerous genes known to be expressed in layer 5 are down-regulated in *Map2k1/2(Mek1/2)* deleted cortices.

*Essential revisions: 1) All three reviewers were concerned about the interpretation of the apparent loss of CTIP^+^ corticospinal neurons. Although there is some evidence for cell death, the number of activated caspase positive neurons is small and the authors cannot rule out the possibility that some neurons have converted from a CTIP^+^, corticospinal fate to callosally projecting fate. Although reduced Fezf2 expression is interpreted as reflecting a loss of these neurons, it could also have been down regulated which may cause neurons to switch their fate. Although it would be preferable to gain mechanistic insight into this question by performing additional experiments, the reviewers agreed that this could probably be addressed through textual changes.*

We thank the reviewers for the insightful comments regarding this important mechanistic question. In the revised manuscript we have provided additional analyses to examine the possibility of conversion of CTIP2^+^ neurons to a callosal phenotype. In Figure 4 of the revised manuscript, we report the proportion of layer 5 neurons that express SATB2, a putative marker of callosal projection neurons. We did not detect an increase in SATB2^+^ neurons as might be expected if CTIP2^+^ neurons in layer 5 switched toward a callosal fate (paragraph two, subheading “Caspase-3 activation in Layer 5 neurons after ERK/MAPK deletion”). In fact the proportion was decreased in motor cortex. These results suggest that cell fate conversion did not occur on a large scale, however, this outcome does not definitively rule out the possibility. Therefore, we have now revised statements regarding the precise mechanism underlying the reduction in CTIP2^+^ neurons (paragraph two, subheading “Failure of CST development and absence of large tufted neurons in layer 5” and paragraph two, subheading “Caspase-3 activation in Layer 5 neurons after ERK/MAPK deletion”), and we note altered expression of master transcription factors potentially involved in conversion, such as *Fezf2* (subheading “Gene expression profiling of ERK/MAPK inactivated cortices”). Further, we provide new references and new discussion of the possibility of altered fate specification (paragraph three, subheading “Specific regulation of layer 5 CST neuron morphology”).

Reviewer #1:

The authors use mouse genetics to study the effects of loss and gain of function of ERK/MAPK kinase signaling in the neocortex and find that both manipulations impair normal formation of the corticospinal tract. In addition, loss of function manipulations has more subtle physiological effects on other classes of pyramidal neurons. Because this signaling pathway has previously been implicated in developmental brain disorders in humans, the present results are interesting for understanding how some of the features of these disorders arise.

*My major concern about the paper in its present state is with respect to the interpretation of the results. The authors find that the cortex is reduced in volume but not in cell number. They then find that CTIP^+^ neurons (that normally form the pyramidal tract) are greatly reduced in motor and somatosensory cortex, but are relatively unaffected in visual cortex (where this same class of neurons does not normally persist in projecting to the spinal cord). The interpretation is that there has been a selective loss the CTIP^+^ neurons. In support of this there is an increase in activation of caspase 3, however the number of neurons with activated caspase 3 is small and the authors cannot rule out the possibility that some neurons have converted from a CTIP^+^, corticospinal fate to callosally projecting fate. I do not think that this is less interesting, or that the authors need to do additional experiments to sort out the relative contributions of cell death and change in cell fate. But they should acknowledge this likely possibility. One additional reason for thinking this likely is that it appears in Figure 2 that the vast majority of the YFP16 neurons are thick-tufted L5 neurons (i.e. the large corticofugal neurons with a tuft in layer 1) whereas in the mutant these neurons take on the morphology (and presumably axonal projections and physiology) of the other major L5 population, the callosal neurons. This transformation occurs in knockout of the transcription factor Fezf2, which is upstream of CTIP2 expression.*

Please see response to Essential revisions above.

Reviewer #2:

[…] The scientific methods, number of replicates, statistically analyses, and overall writing of this manuscript are of high quality. One main weakness in this paper is that all the observed phenotypes can be attributed to poor cell survival, and not other more specific mechanisms such as a decrease in plasticity genes. Also, the direct link between MEK and survival is not addressed. The following issues should be addressed before publication to clarify this manuscript:

*1) The authors show that pERK is upregulated in cortex via Western blot. Is pERK selectively increased in layer 5 neurons in control postnatal mice, and as such, is that why this cell type is dramatically affected by the mutation? This should be examined with immunohistochemistry.*

We now provide new IHC data in Figure 1—figure supplement 1 to assess this important point. We do not find marked differences in levels of phospho-ERK1/2 between layer 5 and other layers at P3 (paragraph one, subheading “Excitatory neuron-specific modification of ERK/MAPK activity”). Thus the selective vulnerability of layer 5 neurons does not appear to be due to preferential recruitment of ERK/MAPK signaling during the neonatal period, although there may be limitations of IHC in detecting such differences.

*2) It is interesting that the layer 5 neurons in the visual cortex are not affected. Is the phenotype activity dependent? This makes sense since visual experience occurs after somatosensory experience. The manuscript would be less descriptive and improved by investigating this issue. One possibility would be to compare layer 5 in the barrel cortex from mice with and without whiskers removed.*

The proposed experiments would certainly help dissect whether ERK/MAPK acts selectively during experience-dependent phases of neuronal development. Indeed we plan to broadly explore the relationship between ERK/MAPK signaling and activity dependent phenomena in a follow-up manuscript.

*3) Is cell death also occurring in 2/3? This should be analyzed with caspase-3 activation assay and added to the manuscript.*

We have included new data in Figure 4 that provide more detailed quantitation of changes in the number of cells that express activated Caspase-3 across all cortical lamina. We did not detect a significant increase in activated Caspase-3 labeling in layer 2/3 in mutant mice (paragraph one, subheading “Caspase-3 activation in Layer 5 neurons after ERK/MAPK deletion”).

*4) The reduction in AP threshold and increased mEPSC amplitude could arise due to smaller soma size, via increased sodium channel density and/or increased input resistance. Please mention this in the Discussion.*

We now mention this in the Discussion in paragraph three, subheading “Global regulation of neuronal excitability”. As the reviewer correctly notes, excitatory neurons lacking ERK/MAPK signaling have both a reduced soma size and an accompanied increase in input resistance. Therefore, we now point out that these properties may influence both AP threshold and mEPSC amplitude.

*5) Explore why mIPSC frequency (in Visual Cortex Layer 2/3) is decreased despite the lack of cre expression in inhibitory neurons. Is this due to the deficiency of ERK/MAPK signaling postsynaptically causing decreased GABAergic synapse number? The authors should investigate this by immunostaining for GABA receptors on 2/3 neurons or inhibitory synapse number by co-localization of a pre and postsynaptic marker around 2/3 cell bodies with immunochemistry.*

We agree with the reviewer that decreased GABAergic synapse number is an interesting possibility. One interpretation of our data is that ERK/MAPK acts as an intermediary in a BDNF-dependent loop that promotes inhibitory synapse formation. We now discuss this possibility and have provided references that support this concept in paragraph two, subheading “Global regulation of neuronal excitability”.

Reviewer #3:

*This is a nice paper examining the role of ERK/MAPK pathway in neocortical development and corticospinal tract (CST) formation. They show that the number of Bcl11b/Ctip2-positive neurons in layer 5 and CST axons length decrease with both, reduced and increased expression of Mek1/2, a mediator of ERK/MAPK pathway. Apart from effects of ERK/MAPK pathway on CST, the authors show that Mek1 deletion, enhanced intrinsic excitability in both layers 2/3 and 5 and created imbalanced synaptic excitation and inhibition.*

*The paper is well written, and the experiments are generally well performed and easy to interpret. I have a few quibbles about the experiments and think some text revisions are in order, but overall this is a nice contribution to the literature that should be well received. Major concern:*

*1) What mediates ERK/MAPK pathway function in layer 5 corticospinal neurons remains unanswered. Moreover, this study contradicts earlier findings by Ozdinler and Macklis, 2006, where they show that IGF1/IGFR1 drives CST formation in vitro and in vivo. In my opinion, the authors failed to take the full advantage of their microarray dataset to address this particular issue. First, their microarray data shows that Bcl11b/Ctip2 expression is maybe slightly decreased, which is consistent with their other data. On the other hand, the expression levels of Fezf2 (also known as Fezl or Zfp312) (PMID: 16157277; PMID: 16314561; PMID: 16284245) and Sox5 (PMID: 18840685; PMID: 18215621), which were previously shown to be necessary for the specification of layer 5 projection neurons and the formation of CST, are dramatically reduced. Also, inactivation of Fezf2 leads to reduced soma size and dendritic arborization of layer 5 subcerebral (corticospinal) neurons (PMID: 16314561).Thus, it is possible that the two transcription factors are involved in mediating the effects of the ERK/MAPK pathway in layer 5 corticospinal neurons. Moreover, the authors speculate in paragraph one, subheading “ERK/MAPK signaling is necessary for corticospinal axon extension”, that the effects are postmitotic as "Nex:Cre is not expressed until neurons are post-mitotic". Actually the SOX5 is expressed only postmitotically/postmigratory in layer 5 projection neurons according to PMID: 18840685, further reinforcing this possibility. Thus, the link to Fezf2 and Sox5 should be explored further as a possible mechanism. One possible experiment would be to overexpress or reduce Fezf2 or Sox5 in layer 5 neuron of Mek1/2cKO or caMEK1, respectively.*

Please see response to Essential revisions above.